# Diminishing seasonality of subtropical water availability in a warmer world dominated by soil moisture–atmosphere feedbacks

Sha Zhou [1,2] ✉, A. Park Williams [3], Benjamin R. Lintner [4], Kirsten L. Findell [5], Trevor F. Keenan [6,7], Yao Zhang [8] & Pierre Gentine [9]

Global warming is expected to cause wet seasons to get wetter and dry seasons to get drier, which would have broad social and ecological implications. However, the extent to which this seasonal paradigm holds over land remains unclear. Here we examine seasonal changes in surface water availability (precipitation minus evaporation, P–E) from CMIP5 and CMIP6 projections. While the P–E seasonal cycle does broadly intensify over much of the land surface, ~20% of land area experiences a diminished seasonal cycle, mostly over subtropical regions and the Amazon. Using land–atmosphere coupling experiments, we demonstrate that 63% of the seasonality reduction is driven by seasonally varying soil moisture (SM) feedbacks on P–E. Declining SM reduces evapotranspiration and modulates circulation to enhance moisture convergence and increase P–E in the dry season but not in the wet season. Our results underscore the importance of SM–atmosphere feedbacks for seasonal water availability changes in a warmer climate.

Changes in surface water availability (defined as precipitation minus evapotranspiration, P–E) over land have widespread consequences for human and natural systems in a warmer climate[1,2]. For example, alterations in the seasonal patterns of precipitation and evapotranspiration may enhance flood and drought risks[3,4], and pose great challenges to local populations, food security, and sustainable management of water resources. Global warming increases water vapor in the atmosphere. This increase is generally expected to amplify the existing spatial as well as seasonal patterns of P–E, leading to wet regions/seasons getting wetter, and dry regions/seasons getting drier, which is referred to as "wet get wetter, dry get drier" (WWDD) mechanism[5–8]. While evidence for the seasonal pattern of WWDD, reflected in both precipitation and P–E, has been found at global and regional scales, mainly over extratropical regions[6,9–11], over some

subtropical dry regions, observations and model projections point to the opposing pattern of seasonal change, with wet seasons becoming drier and dry seasons becoming wetter (WDDW)[9,11]. This unexpected WDDW pattern, if demonstrated to be mechanistically plausible, would have important implications for the reliability of water resources and the sustainability of terrestrial ecosystems, particularly in dry regions. It is therefore crucial to identify the extent to which the WDDW pattern is robust over land and determine the underlying mechanisms involved.

In this work, we examine projected seasonal pattern of P–E changes from the Coupled Model Intercomparison Project Phase 5 (CMIP5)[12] and Phase 6 (CMIP6)[13] and identify the thermodynamic and dynamic mechanisms responsible for seasonal P–E changes using land–atmosphere coupling sensitivity experiments from

[1]State Key Laboratory of Earth Surface Processes and Resource Ecology, Faculty of Geographical Science, Beijing Normal University, Beijing, China. [2]Institute of Land Surface Systems and Sustainable Development, Faculty of Geographical Science, Beijing Normal University, Beijing, China. [3]Department of Geography, University of California, Los Angeles, CA, USA. [4]Department of Environmental Sciences, Rutgers, The State University of New Jersey, New Brunswick, NJ, USA. [5]Geophysical Fluid Dynamics Laboratory, National Oceanic and Atmospheric Administration, Princeton, NJ, USA. [6]Department of Environmental Science, Policy and Management, University of California, Berkeley, CA, USA. [7]Climate and Ecosystem Sciences Division, Lawrence Berkeley National Laboratory, Berkeley, CA, USA. [8]Sino–French Institute for Earth System Science, College of Urban and Environmental Sciences, Peking University, Beijing, China. [9]Department of Earth and Environmental Engineering, Columbia University, New York, NY, USA. ✉e-mail: shazhou21@bnu.edu.cn

CMIP6. We find a robust pattern of increasing dry-season P–E and decreasing wet-season P–E over subtropical regions and the Amazon, which is dominated by seasonally varying soil moisture–atmosphere feedbacks, as drying of the soil reduces evapotranspiration and modulates atmospheric circulation to enhance moisture convergence and increase water availability in the dry season but not in the wet season. These results advance our understanding of seasonal shifts in water availability and underscore the need for more in-depth assessments of hydrological changes over subtropical dry regions.

## Results

### Projected seasonal pattern of water availability changes

We use multi-model simulations from CMIP5 and CMIP6 to investigate seasonal changes in P–E between historical (1971–2000) and future (2071–2100, high-end forcing) periods (Methods). For each model, we define the dry and wet seasons as the three consecutive months with the lowest and highest climatological mean P–E in the historical period, respectively (Supplementary Fig. 1). The spatial patterns of seasonal changes in P–E between CMIP5 and CMIP6, as shown in Fig. 1, are highly correlated ($r > 0.83$). Over many extratropical regions, both CMIP5 and CMIP6 project increasing wet-season P–E and decreasing dry-season P–E, thereby leading to enhanced P–E seasonality (Fig. 1a–f). The seasonality of P–E is also enhanced over tropical Africa and Southeast Asia, mainly due to wet-season P–E increases, with a larger amplitude evident in CMIP6 relative to CMIP5. The enhancement of the annual P–E range is consistent with previous studies indicating that global warming increases water vapor in the atmosphere and amplifies horizontal water vapor transport to strengthen the seasonal cycle of water availability[5,11,14], while changes in atmospheric dynamics may also modify seasonal shifts in P–E[6,9]. However, CMIP5 and CMIP6 also project some regions with wet-season P–E decreases and dry-season P–E increases, mostly over the subtropics and the Amazon. Although the magnitudes of seasonal P–E changes are uncertain and model dependent, the signature of the drying of wet seasons and the wetting of dry seasons (i.e., WDDW) is significant ($p$ value < 0.05, see Methods) for ~20% of global land area (excluding Antarctica and Greenland), leading to reduced seasonality of P–E

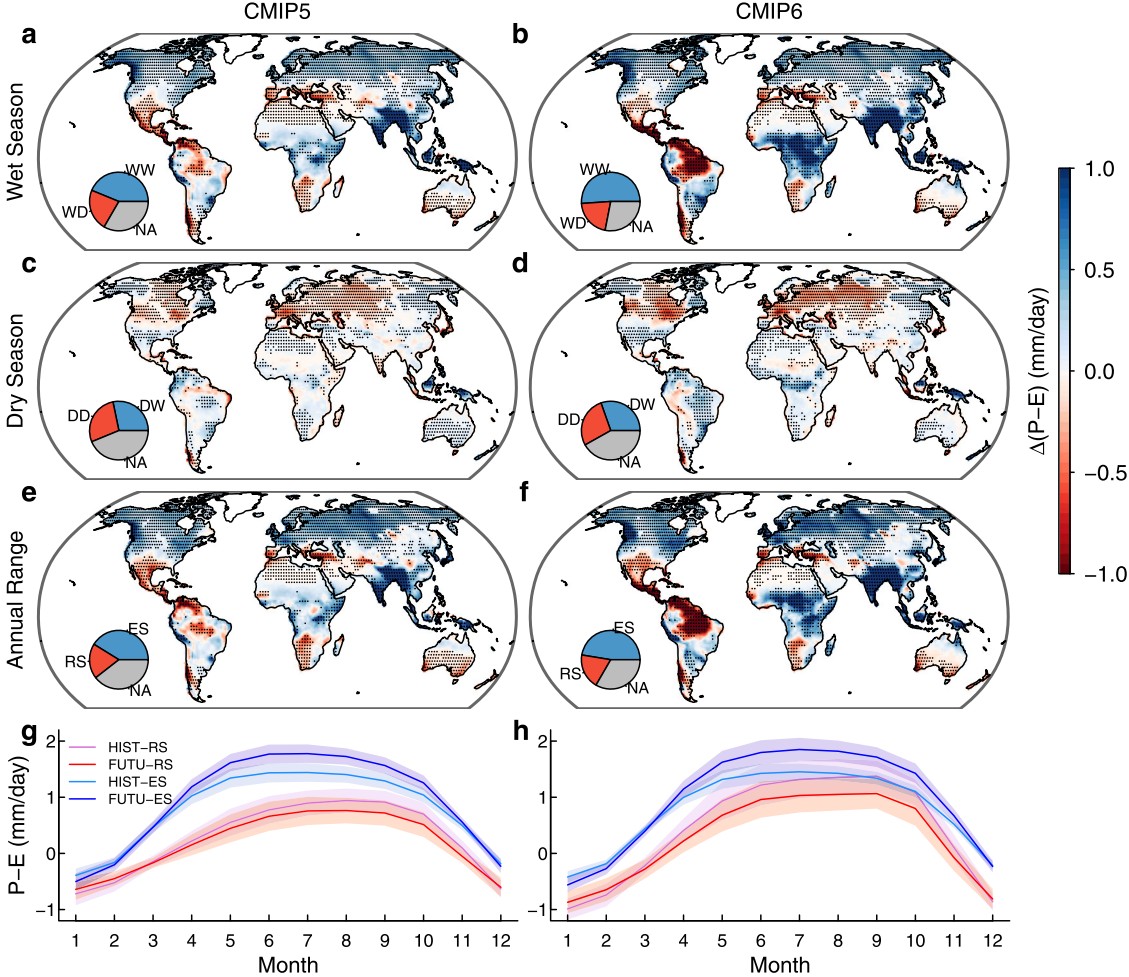

**Fig. 1 | Multi-model mean seasonal changes in water availability in CMIP5 and CMIP6. a–d** Changes in water availability (Δ(P–E)) in the wet season (**a**, **b**) and dry season (**c**, **d**) between historical (1971–2000) and future (2071–2100, RCP8.5 or SSP585) periods (future minus historical). **e, f** The same as (**a–d**) but for changes in the annual range of P–E between wet and dry seasons. The dry/wet season is defined as three consecutive months with lowest/highest mean P–E in the historical period in each model. Stippling denotes regions where the sign of Δ(P–E) is significantly robust ($p$ value < 0.05), i.e., the sign is consistent with the sign of multi-model means (as shown in the figure) for more than 65% of the 35 CMIP5 models and of the 30 CMIP6 models (see Methods). The pie chart insets show proportions of land area with (stippling) and without robust P–E changes. Antarctica and Greenland are excluded. DD (DW) represents dry season showing robust P–E decreases (increases), while WW (WD) represents wet season showing robust P–E increases (decreases). RS (ES) represents reduced (enhanced) seasonality of P–E, assessed as robust decreases (increases) in the annual range of P–E. "NA" represents insignificant changes in P–E ($p$ value > 0.05). **g, h** Mean seasonal cycle of P–E for RS and ES regions in the historical (HIST) and future (FUTU) periods. The sequence of months is organized to start from the first month of dry season. The shading in **g, h** shows the standard deviation of P–E across the assessed models.

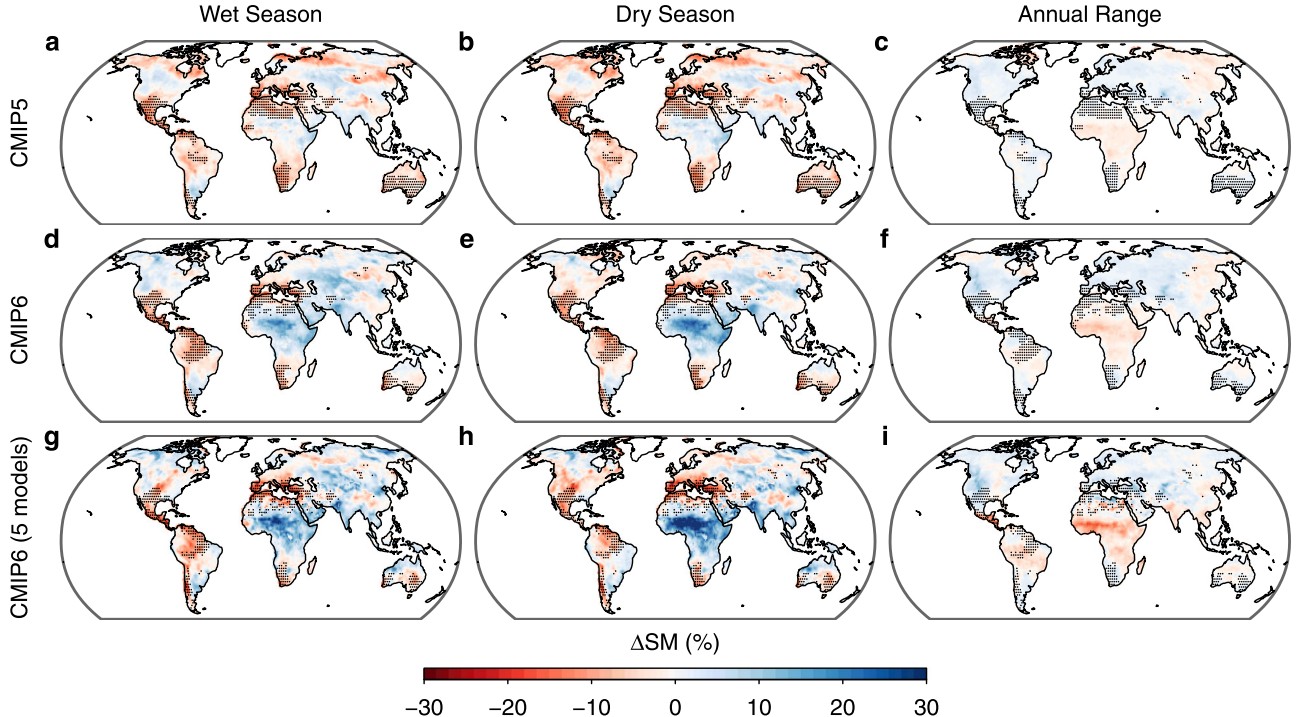

**Fig. 2 | Multi-model mean seasonal changes in soil moisture in CMIP5 and CMIP6.** The percent change in soil moisture (ΔSM) is assessed between 1971–2000 (historical simulation) and 2071–2100 (RCP8.5 or SSP585 simulation) in 35 CMIP5 models (**a**–**c**) and 30 CMIP6 models (**d**–**f**), and between 1980–2000 (historical simulation) and 2080–2100 (SSP585 simulation) in the five CMIP6 models which also participated in the LFMIP-pdLC experiment (**g**–**i**). Stippling denotes reduced seasonality regions in Fig. 1e (35 CMIP5 models, first row), Fig. 1f (30 CMIP6 models, second row), and Fig. 3c (5 CMIP6 models, third row).

(Fig. 1a–f). Further, comparison of seasonal cycles of P-E between historical and future periods also demonstrates opposing seasonal shifts in P-E over the reduced and enhanced seasonality regions (Fig. 1g, h). We have also tested the sensitivity of our results to defining the dry and wet seasons as the three consecutive months with lowest and highest climatological mean P-E in each of the historical and future periods for each model (Supplementary Fig. 1). Overall, while this method yields a smaller percentage (~10% of global land area) of reduced seasonality regions (Supplementary Fig. 2), the spatial patterns of the recalculated seasonal changes of P-E are similar to those in Fig. 1, with spatial correlations of 0.94 and 0.98 for the dry and wet seasons, respectively.

**Mechanisms of seasonal water availability changes**

Long–term P-E changes are driven by both thermodynamic and dynamic processes, which have been widely investigated[15–17]. Global warming, from a thermodynamic perspective, is expected to increase atmospheric water vapor and horizontal moisture transport, favoring increased P-E over wet regions of the tropics and extratropics and reduced P-E over subtropical dry regions[5,17,18]. This thermodynamic effect should also apply for seasonal changes in water availability, with wet seasons getting wetter and dry seasons getting drier[6,9,11]. On the other hand, atmospheric dynamic processes driven by ocean and land region warming (and their contrasts) may also drive P-E change. For example, the potential expansion of the Hadley cell shifts the descending branches poleward and causes subtropical drying[19–22]. This dynamic effect may contribute to the drying of wet seasons, particularly to the extent that associated poleward displacements of storm tracks shift the locus of rain-producing synoptic disturbances poleward, although it is unclear that this should account for the wetting of dry seasons. Overall, the existing thermodynamic and dynamic mechanisms appear to be insufficient to explain the projected WDDW pattern over subtropical regions and the Amazon.

A recent study indicates that soil moisture (SM) also plays an important role in regulating long-term P-E changes[23]. Over subtropical regions and the Amazon, SM is projected to decrease (Fig. 2), which strongly limits evapotranspiration and reduces moisture recycling for precipitation[24,25]. However, by shifting the surface turbulent flux partitioning toward sensible heating, i.e., increasing the surface Bowen ratio, reduced SM may enhance low-level flow convergence and associated moisture convergence, thereby contributing to weaker declines in precipitation than in evapotranspiration, resulting in an increase in P-E and a negative SM feedback on P-E[23]. This implies that increasing dry-season P-E may be associated with local drying of the soil.

To examine whether SM-atmosphere feedbacks can explain the WDDW pattern, we first compare future minus historical SM changes between the wet and dry seasons. In both CMIP5 and CMIP6, we find declining SM in the WDDW regions for both wet and dry seasons, with small inter-seasonal differences (Fig. 2). This indicates that the WDDW pattern is not due to seasonally asymmetric SM changes. On the other hand, it has been demonstrated that the SM limitation on evapotranspiration is stronger under drier conditions[24], and the SM regulation of precipitation is also intrinsically linked to the SM effect on evapotranspiration[26]. We thus hypothesize that the SM effects on evapotranspiration, and hence P-E, may vary seasonally, contributing to the seasonal P-E changes and the WDDW pattern over subtropical regions and the Amazon.

**Seasonally varying soil moisture effects on water availability**

To investigate the hypothesized SM effect on evapotranspiration and seasonal P-E changes, we take advantage of a multi-model CMIP6 ensemble from the Land Feedback Model Intercomparison Project with prescribed Land Conditions (LFMIP-pdLC)[27]. LFMIP-pdLC is identical to the historical and future (SSP585) simulations throughout the simulation period 1980–2100, except that SM is prescribed as the mean seasonal cycle over 1980–2014 from the historical simulation in

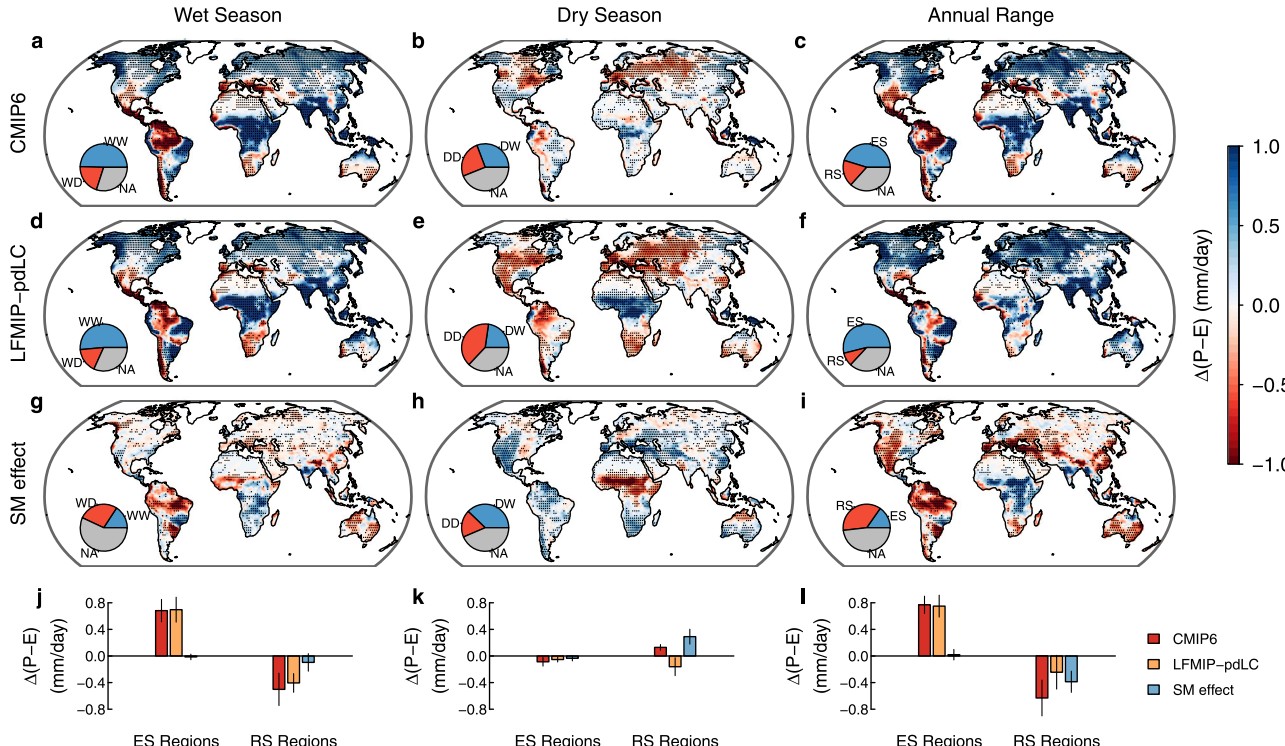

**Fig. 3 | Soil moisture effects on water availability in the wet and dry seasons in CMIP6. a–c** Multi-model mean seasonal changes in water availability (Δ(P-E)) between 1980–2000 (historical simulation) and 2080–2100 (SSP585 simulation) in the five CMIP6 models which participate in the LFMIP-pdLC experiment. **d–f** The same as (**a–c**) but for Δ(P-E) in LFMIP-pdLC (without soil moisture effect). **g–i** Soil moisture (SM) effects on Δ(P-E) assessed as CMIP6 minus LFMIP-pdLC results.

Stippling denotes regions where the sign of Δ(P-E) is consistent with the sign of multi-model means (as shown in the figure) for at least four of the five models. The pie chart insets show proportions of land area with (stippling) and without robust P-E changes, similar to those in Fig. 1. **j–l** Total area-weighted Δ(P-E) in enhanced seasonality (ES) regions and reduced seasonality (RS) regions in CMIP6 (**c**). The error bars show the standard deviation of Δ(P-E) across the five models.

each of the five participating models (CESM2, CNRM-CM6-1, EC-Earth3, IPSL-CM6A-LR, MPI-ESM1-2-LR; see Methods). We isolate the SM effect on seasonal P-E changes between the historical (1980–2000) and future (2080–2100) periods as the five-model mean difference between CMIP6 and LFMIP-pdLC.

Our comparison of the CMIP6 and LFMIP-pdLC simulations reveals distinct SM effects on evapotranspiration and P-E between wet and dry seasons (Fig. 3 and Supplementary Fig. 3). As expected, the SM limitation on evapotranspiration is stronger in the dry season than the wet season over subtropical regions and the Amazon (Supplementary Fig. 3d, e). In the wet season, P-E changes in CMIP6 and LFMIP-pdLC are comparable over most land area (Fig. 3a, d), and the SM effect on P-E changes is relatively small (Fig. 3g), especially in the Northern Hemisphere, compared to P-E changes induced by other processes, such as anthropogenic climate change, in LFMIP-pdLC (collectively, we term these the non-SM effect). The drying of the wet season over subtropical regions and the Amazon is therefore mainly caused by the non-SM effect (Fig. 3a, d, g), as anthropogenic warming reduces precipitation but enhances evapotranspiration with prescribed SM in LFMIP-pdLC (Supplementary Fig. 4a, d). However, the SM effect is opposite, and of similar magnitude, to the non-SM effect on P-E changes in the dry season (Fig. 3e, h). Over subtropical regions and the Amazon, the non-SM effect reduces P-E and SM in both seasons (Figs. 3d, e, 2g, h). SM drying and associated land-atmosphere processes strongly increase dry-season P-E, which cancels out P-E reductions induced by the non-SM effect, resulting in an increase in the net P-E and the wetting of the dry season (Fig. 3b, e, h). Although the magnitude of SM drying is similar in both seasons (Fig. 2g, h), the negative SM feedback on P-E is much weaker in the wet season than the dry season (Fig. 3g, h), contributing to the WDDW pattern and reduced seasonality of P-E over 18% of land area in CMIP6

coupled simulations (Fig. 3c). Without the SM effect in LFMIP-pdLC, only 9% of land area experiences reduced seasonality of P-E (Fig. 3f). In the reduced seasonality regions, 80% of wet-season P-E reductions (−0.50 ± 0.24 mm/day) are caused by the non-SM effect, with the remaining 20% from the SM effect. In contrast, the positive SM effect on dry-season P-E (0.29 ± 0.11 mm/day) is roughly twice the magnitude of non-SM induced P-E reductions (−0.16 ± 0.13 mm/day) (Fig. 3j, k). Overall, the distinct SM effects between wet and dry seasons (−0.39 ± 0.16 mm/day) are responsible for 63% of the reduced annual range of P-E, while the non-SM effect (−0.24 ± 0.25 mm/day) accounts for the remaining 37% (Fig. 3l).

The above SM effect on seasonal P-E changes highlights the seasonally varying nature of SM-(P-E) feedbacks, but also includes the effect of changes in SM climatology (i.e., mean seasonal cycle of SM, Fig. 2g–i). To directly compare the SM-(P-E) feedbacks between dry and wet seasons, we apply an empirical statistical method (Methods) to CMIP6 models and observationally constrained reanalysis products (Modern-Era Retrospective analysis for Research and Applications (MERRA-2)[28] and European Center for Medium-Range Weather Forecasts (ERA5)). The five CMIP6 models and two reanalysis products consistently show strong negative SM-(P-E) feedbacks over subtropical regions in the dry season, when the positive SM effect on evapotranspiration exceeds that on precipitation (Fig. 4a–c, g–i). However, the negative SM-(P-E) feedback is weak in the wet season, when SM limitation on evapotranspiration is negligible (Fig. 4d–f, j–l). These results support the conclusion from Fig. 3 of the presence in subtropical regions of a strong effect of SM drying on P-E increases in the dry season but a weak SM effect in the wet season. Over the Amazon, the SM-(P-E) feedbacks are mostly positive (though not statistically significant) in the wet season and negative in the dry season (Fig. 4a, d, g, j), consistent with wet-season P-E decreases and dry-season P-E

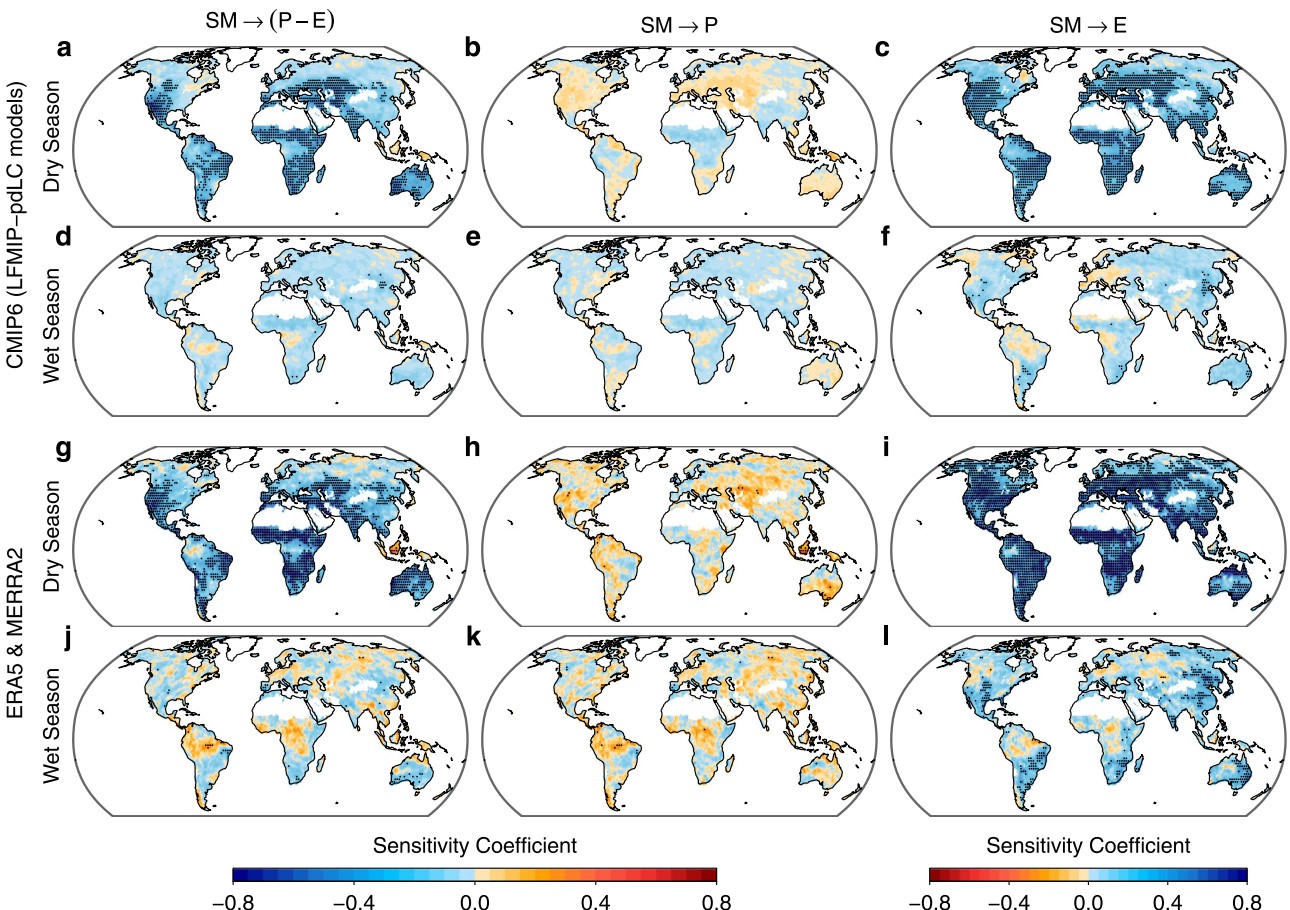

**Fig. 4 | Soil moisture feedbacks on water availability in the wet and dry seasons.** **a–f** Mean sensitivity coefficients for soil moisture (SM)→precipitation minus evapotranspiration (P-E), SM→P, and SM→E identified based on historical and SSP585 simulations (1980–2100) of the five CMIP6 models which participate in the LFMIP-pdLC experiment. **g–l** The same as (**a–f**), but for mean sensitivity coefficients from reanalysis products ERA5 (1979–2019) and MERRA-2 (1980–2019) (**d–f**). The sensitivity coefficient for X→Y denotes the partial derivative of Y in the wet/dry seasons to X in the prior month. In each model/reanalysis product, the seasonal cycles and long-term trends in X and Y are removed and the remaining variations of X and Y are standardized in the wet and dry seasons individually. Stippling denotes regions where the sensitivity coefficient is significant at the 95% level according to a bootstrap test and the sign of the sensitivity coefficient is consistent with the sign of multi-model means (as shown in the figure) in all models/reanalysis products.

increases in response to SM drying (Fig. 3g, h). Based on this empirical assessment, we again conclude that seasonally varying SM effects dominate the reduced seasonality of P-E over subtropical regions and the Amazon.

### Mechanisms of the SM-atmosphere feedbacks in the wet and dry seasons

For further mechanistic insight, we perform an atmospheric moisture budget diagnosis of the SM effect between the wet and dry seasons in CMIP6 and LFMIP-pdLC. In the reduced seasonality regions, SM drying exerts seasonally varying effects on the atmospheric moisture budget (Fig. 5b, d, f). Dry-season evapotranspiration reductions are largely offset by increased moisture convergence (or decreased moisture divergence due to reduced supply of water vapor through evapotranspiration), resulting in only small decreases in precipitation (Fig. 5d). In the wet season, the SM effect reduces both evapotranspiration and moisture convergence, and induces large reductions in precipitation (Fig. 5b). However, the SM effect on the atmospheric moisture budget is small over the enhanced seasonality regions (Fig. 5a, c, e), consistent with the weak SM effects on evapotranspiration, precipitation, and P-E from reanalysis products (Figs. 3, 4).

At the seasonal scale, the change in atmospheric moisture storage is relatively small; thus P-E approximately equals moisture convergence. The SM effects on P-E and moisture convergence are therefore spatially congruent (Fig. 3g–i and Supplementary Fig. 3g–i), with spatial correlations greater than 0.85 in both seasons. By decomposing moisture convergence changes into thermodynamic and dynamic terms (Methods), we find the difference in the SM effects on moisture convergence (and P-E) between wet and dry seasons is mainly associated with the dynamic effect in the reduced seasonality regions (Fig. 5b, d, f). In the dry season, the drying of land surface reduces evaporative cooling and strongly enhances land warming, thereby reducing air pressure over subtropical land and the Amazon relative to the ocean and many moist land regions (Supplementary Fig. 5b, d, f). The strengthened surface pressure gradient favors low-level flow convergence and vertical ascent (represented by negative pressure velocity) throughout the troposphere over subtropical land and the Amazon (Fig. 5h and Supplementary Fig. 6e–h). However, the SM effects on land surface warming and atmospheric vertical motion are weaker in the wet season, with the negative pressure velocity slightly enhanced in the low-level troposphere but suppressed in the mid- and high-level troposphere (Fig. 5g and Supplementary Fig. 6a–d). Enhanced/suppressed vertical ascent is associated with enhanced/suppressed net moisture flux from the ocean to land, as inferred by the strong spatial correlation between changes in the negative pressure velocity and changes in moisture convergence (or P-E) over land (Supplementary Fig. 6i–l).

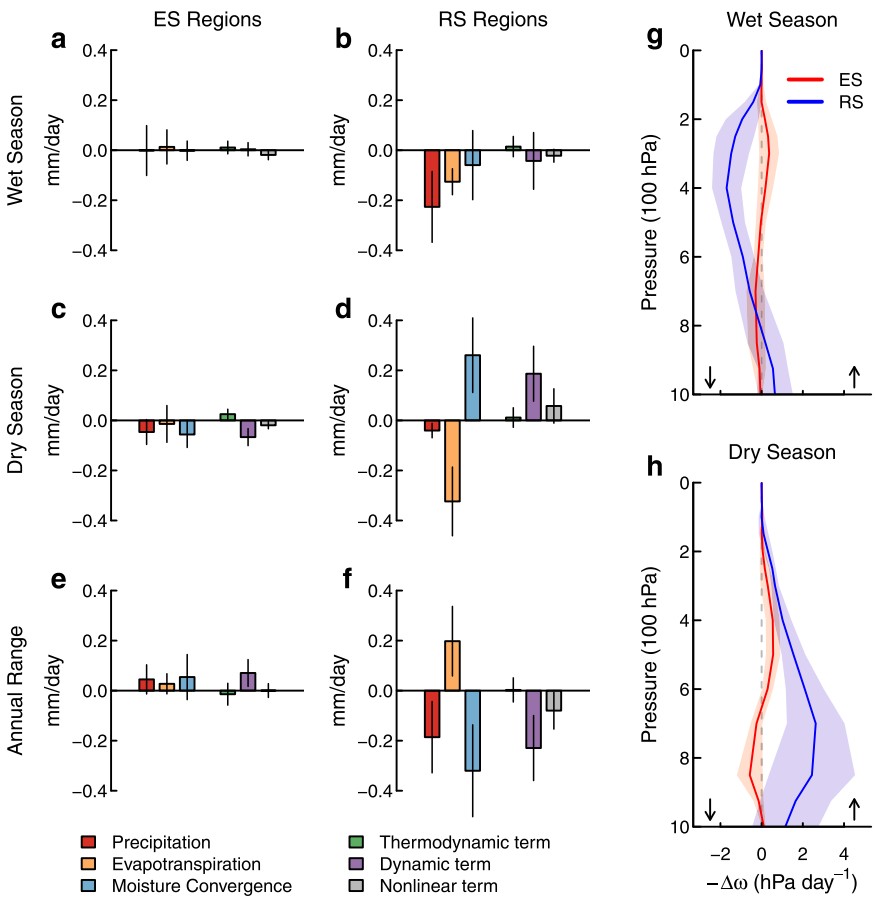

**Fig. 5 | Mechanisms of the soil moisture effects on water availability in the wet and dry seasons. a–f** Soil moisture (SM) effects on changes in precipitation, evapotranspiration, moisture convergence, and the thermodynamic, dynamic, and nonlinear terms of moisture convergence between 1980–2000 and 2080–2100 in enhanced seasonality (ES) regions and reduced seasonality (RS) regions shown in Fig. 3c. The SM effect is assessed as the five-model mean difference between CMIP6 and LFMIP-pdLC (similar to Fig. 3g–i). **g, h** SM effects on negative pressure velocity (−△ω) over ES and RS regions. The error bars in **a–f** and the shading in **g, h** show the standard deviation of each variable across the five models. The upward and downward arrows in **g, h** show vertical ascent and descent, respectively.

## Discussion

This study demonstrates a robust pattern of wet seasons getting drier and dry seasons getting wetter, from the perspective of P-E, over a significant fraction of subtropical land regions and the Amazon in both CMIP5 and CMIP6 projections. Further analyses of CMIP6 land-atmosphere coupling sensitivity experiments provide evidence that the drying of the wet season is mainly caused by anthropogenic climate change, while the wetting of the dry season in terms of increased P-E is driven by the negative SM feedback on P-E. We have also found that the reduced seasonality of P-E is predominantly caused by the seasonally varying SM effects on P-E, while anthropogenic climate change reduces P-E in both dry and wet seasons, and the net effect on P-E seasonality is small over the subtropics and Amazon. The resultant drying of the soil in turn reduces evapotranspiration and recycling of evaporated moisture for subsequent precipitation[24,25]. Reduced evaporative cooling further amplifies land surface warming, and the associated land-ocean warming contrast strengthens surface pressure differences between ocean and land, which drives anomalous ocean-to-land moisture transport and enhances moisture convergence over land. Such an increase in SM-induced moisture convergence offsets the decrease in precipitation driven by reduced moisture recycling of evapotranspiration, resulting in a more muted precipitation response than the evapotranspiration response to SM drying and a negative SM feedback on P-E. The SM limitation on evapotranspiration and associated SM-atmosphere feedbacks, especially those related to atmospheric dynamics, are strong in the dry season but weak in the wet season, contributing to large P-E increases in the dry season and slight

P-E decreases in the wet season. The seasonally varying nature of SM-atmosphere feedbacks therefore leads to reduced seasonality of P-E over subtropical regions and the Amazon.

While the SM effect on long-term P-E changes in subtropical dry regions has been recognized[23], our modeling and empirical assessments further show that the negative SM-(P-E) feedback occurs mainly in the dry season. The negative SM-(P-E) feedback caused by the SM regulation of atmospheric dynamics and moisture convergence is also expected from observational evidence that the SM limitation on evapotranspiration is strong in the dry season, when precipitation is relatively low and cannot decrease as much as evapotranspiration could in response to SM drying[24]. On the other hand, reduced evapotranspiration leads to reduced terrestrial supply of water vapor for moisture divergence and therefore curbs the reduction of P-E in the dry season. The projected SM declines combined with the negative SM-(P-E) feedback explains the positive SM effect on dry-season P-E increases isolated using modeling experiments over subtropical regions and the Amazon. While the negative sign of the SM-(P-E) feedback is supported by both reanalysis products and CMIP6 models, the feedback strength and the magnitude of SM-induced P-E changes vary across models and products (Figs. 3, 4), which may result from uncertainties in the representation of SM-atmosphere feedbacks.

Seasonal variations in P-E are closely related to extreme hydro-climate conditions, such as droughts and floods, which are projected to increase in many regions under climate change[3,4,29]. Future drought risks and associated carbon loss are likely to be especially strong over the subtropics and the Amazon[29,30]. Increased dry-season P-E due to

SM feedbacks may somewhat attenuate the potential increase of drought risk expected from the thermodynamic hydrological changes. In the absence of the negative SM feedbacks, the projected extreme hydroclimate events would likely become more frequent and more extreme than coupled climate projections[23], which would reduce the capacity of terrestrial ecosystems to absorb $CO_2$ and mitigate climate change in the future.

Our findings underscore the importance of soil moisture-atmosphere feedbacks in modulating seasonal water availability changes. In particular, while warming-driven oceanic and atmospheric changes point to declining dry-season water availability in the future, soil moisture feedbacks can be viewed as offsetting the decline over subtropical dry regions that would be realized in the absence of such feedbacks. Given the widespread opposite effects of soil moisture and other climate factors on water availability in the dry season (Fig. 3e, h), soil moisture-atmosphere feedbacks may alleviate the negative climate change implications for regional water resources management. It is worth noting that while soil moisture-atmosphere feedbacks lead to increases in surface water availability in the dry season, reduced soil moisture itself and associated declining evapotranspiration also indicate an overall drying trend of the land surface system driven by climate change over subtropical dry regions and the Amazon. As the soil moisture feedback on surface water availability is stronger in drier conditions, human activities like irrigation may weaken the negative soil moisture feedback and magnify water scarcity in dry regions, but the scale at which this effect might be detected need to be identified. Our study enables a mechanistic understanding of the role of soil moisture-atmosphere feedbacks in regulating the seasonal pattern of water availability in coupled climate models, while a more in-depth assessment of regional hydrological changes and associated hydroclimate extremes and vegetation activities based on observations and model projections is needed. As subtropical ecosystems are among the most vulnerable to climate change[31], it is crucial to continue to assess seasonal variations in the freshwater resources and terrestrial ecosystems and refine projections of the coupled climate-hydrology-ecosystem to promote effective conservation actions.

## Methods

### CMIP5 and CMIP6 model simulations

We used output from 35 CMIP5 models (Supplementary Table 1) and 30 CMIP6 models (Supplementary Table 2) covering the historical (1971–2000) and future (2071–2100) periods. The high-end forcing scenarios (RCP8.5 in CMIP5 and SSP585 in CMIP6) were used in future simulations. We used these models because they provide the monthly total soil moisture content ("mrso"), precipitation ("pr"), and latent heat flux ("hfls") as required for our analyses. For each model, one ensemble member was used (see Supplementary Tables 1, 2 for details). We calculated evapotranspiration from latent heat flux which was available in more CMIP6 models than evapotranspiration ("evspsbl"), and obtained precipitation minus evapotranspiration (P-E) in each model. In CMIP5 and CMIP6, we defined the wet and dry seasons as the three consecutive months with the highest and lowest climatological mean P-E, respectively, using data from the historical simulation (1971–2000) in each model. Multi-model mean seasonal changes in soil moisture, evapotranspiration, precipitation, and P-E between the historical and future periods were calculated.

### Soil moisture-atmosphere feedback experiments

We used a new multi-model experiment from the Land Feedback Model Intercomparison Project with prescribed Land Conditions (LFMIP-pdLC), which was designed to assess land surface feedbacks on climate change in CMIP6[27]. LFMIP-pdLC performed transient coupled atmosphere-ocean simulations driven by the same forcing, including sea surface temperature and sea ice, from corresponding CMIP6 simulations (the historical simulation during 1980–2014 and

the SSP585 scenario during 2015–2100), except that soil moisture (SM) was prescribed as the mean seasonal cycle of 1980–2014 from the historical simulation in each model (https://wiki.c2sm.ethz.ch/LS3MIP/Tier1Experiments). LFMIP-pdLC is currently available for five models (CESM2, CNRM-CM6-1, EC-Earth3, IPSL-CM6A-LR, MPI-ESM1-2-LR). Comparing the fully coupled historical and future (SSP585) simulations in CMIP6 (expressed as CMIP6 simulations below) and LFMIP-pdLC, we could assess the SM effect on P-E in each model.

We used monthly total soil moisture content, precipitation, and latent heat flux from these simulations. We assessed seasonal changes in the variables (SM, precipitation, evapotranspiration, and P-E) between 1980–2000 and 2080–2100 in CMIP6 and LFMIP-pdLC, the latter only covers 1980–2100. Correspondingly, the wet and dry seasons are defined using historical data from 1980 to 2000 in CMIP6. Although long-term hydrological changes are usually assessed at 30-year time scales (Fig. 1), we compared seasonal changes in precipitation, evapotranspiration, and P-E between 1971–2000 and 2071–2100 and between 1980–2000 and 2080–2100 in CMIP6, and found the results are very close (Supplementary Fig. 7).

We used pressure level data from CMIP6 and LFMIP-pdLC to assess the thermodynamic and dynamic mechanisms of the SM effect on seasonal P-E changes. We used monthly specific humidity, eastward and northward wind on pressure levels, and surface pressure to calculate moisture convergence and decompose it into thermodynamic and dynamic terms (see "Moisture budget decomposition"). We also used near-surface (2 m) air temperature, pressure velocity on pressure levels, and sea level pressure in the five models that participate in both CMIP6 and LFMIP-pdLC for the mechanistic analyses.

### Reanalysis datasets

To support the modeling feedback analyses, we identified the SM-atmosphere feedbacks using two state-of-the-art reanalysis products: the Modern-Era Retrospective analysis for Research and Applications, version 2 (MERRA-2)[28] and the European Center for Medium-Range Weather Forecasts (ERA5). MERRA-2 and ERA5 are constrained by in situ and satellite remote sensing observations, and reasonably capture the relationship between SM and P-E[23]. We used monthly root-zone (0–100 cm) SM, precipitation, evapotranspiration from MERRA-2 (1980–2019) and ERA5 (1979–2019) to assess the SM-atmosphere feedbacks in the wet and dry seasons (see "Soil moisture-atmosphere feedbacks").

### Robustness of the seasonal changes in P-E

We made use of ensembles of CMIP5 and CMIP6 models to test the robustness of the seasonal changes in P-E ($\Delta$(P-E)). As the sign and magnitude of $\Delta$(P-E) vary across models, we used the multi-model mean $\Delta$(P-E) as the best estimate and tested whether the sign of multi-model means is statistically robust. For each grid cell, if the multi-model mean $\Delta$(P-E) is positive, we tested the following hypothesis:

1. The null hypothesis is that the sign of $\Delta$(P-E) is random, so the probability of a positive $\Delta$(P-E) is 0.5 ($p = 0.5$);
2. The alternative hypothesis is that $p > 0.5$;
3. To test the null hypothesis, we construct a test statistic: the number of models of all models ($n$) that show positive $\Delta$(P-E);
4. As the sign of $\Delta$(P-E) is independent across different models, the number of models with positive $\Delta$(P-E) follows the binomial distribution, and the probability of positive $\Delta$(P-E) simulated in exactly $m$ models is given by

$$P_m = \frac{n!}{m!(n-m)!}p^m q^{n-m}, p = 0.5, q = 0.5 \qquad (1)$$

5. According to the probability density function of binomial distribution, if positive $\Delta$(P-E) occurs in 23 of 35 (66%) CMIP5 models or 20 of 30 (67%) CMIP6 models, we can reject the null hypothesis

($p$ value < 0.05), and the positive sign of multi-model mean $\Delta$(P-E) is significantly robust.

$$P_{CMIP5}(m \geq 23) = \sum_{m=23}^{35} \frac{35!}{m!(35-m)!} 0.5^m \times 0.5^{35-m} < 0.05 \quad (2)$$

$$P_{CMIP6}(m \geq 20) = \sum_{m=20}^{30} \frac{30!}{m!(30-m)!} 0.5^m \times 0.5^{30-m} < 0.05 \quad (3)$$

The similar hypothesis testing was also applied for grid cells with negative multi-model mean $\Delta$(P-E). In general, if the sign of $\Delta$(P-E) is consistent with the sign of multi-model means for more than 65% of the 35 CMIP5 models and of the 30 CMIP6 models, the sign of multi-model mean $\Delta$(P-E) is deemed to be statistically significant at the 95% confidence level.

### Soil moisture-atmosphere feedbacks

We applied an empirical statistical method to assess the SM-(P-E) feedbacks in the wet and dry seasons using the two reanalysis products and CMIP6 models. This method establishes a multiple linear regression model between P-E and one-month lagged SM to identify the sign and strength of the SM-(P-E) feedback[23]. As the SM effect on P-E may persist for weeks to months, the regression model between P-E and 1-month lagged SM therefore can isolate the SM feedback on P-E from the direct P-E effect on SM[23]. In the regression model, the multi-year mean seasonal cycles and the linear trends of SM and P-E are removed to focus on the feedback of SM variations on P-E variations. The regression model also takes into account the prior-month P-E to overcome the potential effect of P-E autocorrelation.

$$(P-E)_d(t) = n_0 + n_1 \cdot SM_d(t-1) + n_2 \cdot (P-E)_d(t-1) \quad (4)$$

where the subscript $d$ indicates that the seasonal cycle and linear trend of the variable are removed, and the indicator $t$ represents monthly steps in the wet or dry season. The regression coefficient $n_1$ represents the partial derivative of P-E variations to SM variations in the prior-month $\left(\frac{\partial (P-E)_d(t)}{\partial SM_d(t-1)}\right)$, and was used to capture the SM feedback on P-E in the wet and dry seasons.

We identified the SM feedback on P-E as the standardized $n_1$, or sensitivity coefficient for SM→(P-E), which corresponds to standardized $(P-E)_d$ and $SM_d$ of zero mean and unit variance in the wet or dry season. In this way, we could better compare the SM-(P-E) feedback between wet and dry seasons and across different regions/datasets/models. Alternatively, we standardized the entire time series of $(P-E)_d$ and $SM_d$ to zero mean and unit variance and then applied the regression model to obtain the regression coefficient $n_1$. The spatial patterns of the identified sensitivity coefficients for SM→(P-E) from the two standardization methods are identical (Fig. 4 and Supplementary Fig. 8).

We used a bootstrap test to determine the significance of the sensitivity coefficients in case the identified SM-(P-E) feedback may be sensitive to natural variability. In the bootstrap analysis, the time series of the variables are randomly resampled to perform the multiple linear regression and obtain the 95% confidence intervals of the sensitivity coefficients for the wet and dry seasons. According to the bootstrap confidence intervals, the sensitivity coefficients are deemed statistically significant if the 95% confidence intervals do not overlap with zero. The multiple linear regression method and the bootstrap test were also used to obtain the sensitivity coefficients for the SM effects on evapotranspiration and precipitation.

### Moisture budget decomposition

According to atmospheric moisture budget, P-E equals to moisture convergence (MC), which is defined as the negative divergence of vertically integrated moisture flux over the pressure ($p$) from the top of the atmosphere ($p = 0$) to the surface ($p = p_s$).

$$P - E = MC \quad (5)$$

$$MC = -\frac{1}{\rho_w g} \nabla \cdot \int_0^{p_s} (\boldsymbol{u}q) dp \quad (6)$$

where $\rho_w$ is the density of water, $g$ is the acceleration due to gravity, $\nabla$ is the horizontal divergence operator, $\boldsymbol{u}$ is the horizontal vector wind, and $q$ is specific humidity.

As we focused on climatological seasonal changes of P-E, we used monthly data to approximately calculate MC differences between the historical and future periods.

$$MC \approx -\frac{1}{\rho_w g} \nabla \cdot \int_0^{p_s} (\bar{\boldsymbol{u}} \cdot \bar{q}) dp \quad (7)$$

where overbars indicate monthly mean values. The change in MC ($\triangle MC$) can be decomposed as

$$\triangle MC \approx -\frac{1}{\rho_w g} \nabla \cdot \int_0^{p_s} (\bar{\boldsymbol{u}}_0 \cdot \triangle \bar{q} + \bar{q}_0 \cdot \triangle \bar{\boldsymbol{u}} + \triangle \bar{\boldsymbol{u}} \cdot \triangle \bar{q}) dp \quad (8)$$

where the subscript 0 represents the historical period, and the delta operator represents changes from the historical to future periods. On the right side of Eq. (8), $\triangle MC$ is decomposed into a thermodynamic term due to specific humidity changes ($-\frac{1}{\rho_w g} \nabla \cdot \int_0^{p_s} (\bar{\boldsymbol{u}}_0 \cdot \triangle \bar{q}) dp$), a dynamic term due to horizontal wind changes ($-\frac{1}{\rho_w g} \nabla \cdot \int_0^{p_s} (\bar{q}_0 \cdot \triangle \bar{\boldsymbol{u}}) dp$), and a nonlinear term due to the product of specific humidity and wind changes ($-\frac{1}{\rho_w g} \nabla \cdot \int_0^{p_s} (\triangle \bar{\boldsymbol{u}} \cdot \triangle \bar{q}) dp$)[15,23,32].

We used monthly specific humidity and zonal and meridional wind velocity on pressure levels, and surface pressure from CMIP6 and LFMIP-pdLC to calculate the change in MC between the historical (1980–2000) and future (2080–2100) periods, and its three components (Eq. 8) in the wet and dry seasons. The SM effects on MC changes and the thermodynamic, dynamic, and nonlinear terms were calculated as their differences between CMIP6 and LFMIP-pdLC. As the dynamic term ($-\frac{1}{\rho_w g} \nabla \cdot \int_0^{p_s} (\bar{q}_0 \cdot \triangle \bar{\boldsymbol{u}}) dp$) is approximately equal to $-\frac{1}{\rho_w g} \int_0^{p_s} (\triangle \bar{\omega} \partial q / \partial p) dp$, where $\omega$ is the pressure vertical velocity, according to the mass continuity equation[33], we identify the dynamic effect by analyzing the SM effects on both the horizontal wind velocity and the pressure vertical velocity throughout the troposphere.

## Data availability

All data used in this study are available online. The CMIP5 model simulations were downloaded from https://esgf-node.llnl.gov/search/cmip5/, and the CMIP6 (including LFMIP-pdLC) model simulations are available from https://esgf-node.llnl.gov/search/cmip6/. The ERA5 reanalysis data are from https://www.ecmwf.int/en/forecasts/datasets/archive-datasets/reanalysis-datasets/era5. The MERRA-2 reanalysis data are from https://gmao.gsfc.nasa.gov/reanalysis/MERRA-2/data_access/.

## Code availability

The R code used for modeling and reanalysis data analyses is publicly available (https://doi.org/10.5281/zenodo.6802965).

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

## Acknowledgements

We acknowledge the World Climate Research Program's Working Group on Coupled Modeling, which is responsible for CMIP, and we thank the climate modeling groups (listed in Supplementary Tables 1, 2 of this paper) and the LFMIP group for producing and making available their model output. For CMIP the U.S. Department of Energy's Program for Climate Model Diagnosis and Intercomparison provides coordinating support and led development of software infrastructure in partnership with the Global Organization for Earth System Science Portals. S.Z. acknowledges the NSFC Excellent Young Scientists Fund (Overseas), the Second Tibetan Plateau Scientific Expedition and Research Program (2019QZKK0405), and the Fundamental Research Funds for the Central Universities. P.G. acknowledges support from NASA ROSES Terrestrial hydrology (NNH17ZDA00IN-THP) and NOAA MAPP NA17OAR4310127. A.P.W. acknowledges support from the NASA Modeling, Analysis, and Prediction (MAP) program (NASA 80NSSC17K0265). T.F.K. acknowledges support from the RUBISCO SFA, which is sponsored by the Regional and Global Model Analysis (RGMA) Program in the Climate and Environmental Sciences Division (CESD) of the Office of Biological and Environmental Research (BER) in the U.S. Department of Energy Office of Science, and additional support from a DOE Early Career Research Program award (DE-SC0021023).

## Author contributions

S.Z. conceived and designed the study. S.Z. processed model simulations and reanalysis data. S.Z., A.P.W., B.R.L., K.L.F., T.F.K., Y.Z., and P.G. contributed to data analysis and interpretation. S.Z. wrote the paper. A.P.W., B.R.L., K.L.F., T.F.K., Y.Z., and P.G. edited the paper.

## Competing interests

The authors declare no competing interests.
