## [Peer Review File · Nature Communications]

Diminishing seasonality of subtropical water availability in a warmer world dominated by soil moisture-atmosphere feedbacksREVIEWER COMMENTS

Reviewer #1 (Remarks to the Author):

The authors analyze CMIP5 and CMIP6 simulations in terms of their changes in P-E. The core result is an association between areas of “wet get drier, dry get wetter” predominantly over subtropical land areas and land-atmosphere feedbacks. I believe major revisions would greatly improve the quality of this study, contributing to the robustness and the understanding of the results.

Major comments:

L78-83: Determining dry and wet seasons based on the extremes in consecutive 3-month precipitation totals (rather than a static set of 3 months as is done here) accounts for potential shifts in seasons in a changing climate. This is a more “honest” approach, and the smaller percentage of reduced seasonality found by this method (results currently in the supplement) is indicative of this – nature does not know of calendars. I think this should be the default way to calculate the changes, and the version fixed on the historical period seasons should be supplemental. Regardless of whether the change is made, there should be more exploration of this contrast in the paper. At a minimum, I suggest two additions: (1) a figure showing how seasons have shifted, which would require a 4-panel plot of: 2 12-color (for JFM, FMA, MAM, etc.) maps of the consensus (median?) wet and dry seasons; and 2 maps of the average shift in months from historical to future; (2) As a companion to Figs 1 and S1, produce another parallel calculation based on using seasons defined based on the future period – this would close the assessment of the sensitivity of the results to the definitions of wet and dry seasons. I suspect the smallest changes will be when shifts in seasons are accounted for, rather than misdefining the seasons in one or the other set of experiments.

Specific comments:

L70: Kumar et al (2014) is cited, but even more germane to this study is Kumar et al. (2015; [https://doi.org/ 10.1002/2015GL066858](https://doi.org/10.1002/2015GL066858)).

Much of what it is Fig S2 should be in the main manuscript, as it is crucial to explaining the results shown in Fig 3. I think the contents of Fig 4 are largely intuitive and less crucial than Fig S2 – Fig 4 could be moved to the supplement as supporting material.

L229 & 260-261: What about a change in soil moisture leads to the changes in winds/advection? It must be though changes in atmospheric heating (changing temperatures hypsometrically affecting thickness, geopotential) via soil moisture controls on surface heating. The flow changes in the subtropics must still be largely geostrophic, and not pressure gradient as conjectured at L260. Again, this can be verified with the data in hand – a streamfunction calculation would quantify this nicely.

Fig 5: Panels g and h: I did not catch at first the minus sign in front of omega until I had looked at the figure several times. It would be helpful to draw vertical arrows next to the pressure levels that show the biggest departure from 0 in the RS cases, just to make it very clear which is associated with ascent and descent.

L258: The speculation about recycling can be checked. It would be straightforward to follow one of the methodologies of bulk moisture transport (e.g., Brubaker et al. 1993; [https://doi.org/10.1175/1520-0442\(1993\)006<1077:EOCPR>2.0.CO;2](https://doi.org/10.1175/1520-0442(1993)006<1077:EOCPR>2.0.CO;2), or alternatively Trenberth 1999; [https://doi.org/https://doi.org/10.1175/1520-0442\(1999\)012<1368:AMRROA>2.0.CO;2](https://doi.org/https://doi.org/10.1175/1520-0442(1999)012<1368:AMRROA>2.0.CO;2)) which are appropriate to monthly model output.

L283-290: The results also imply that irrigation in dry areas would compound aridity – something already seen observationally in places like the upper Ganges and Indus valleys. While speculating on the broader impacts of your results, this could be mentioned.

Regarding use of reanalysis datasets: data assimilation violates conservation and budget closure (not a problem with the free-running CMIP models). This affects the divergent part of flow and transport more than the rotational part, which means moisture flux divergence is less reliable in reanalysis budgets than transport. This is also why reanalysis precipitation biases remain poor, as they are strongly connected to moisture flux convergence. Furthermore, surface fluxes are completely unconstrained by observations. So be wary of putting too much trust in this aspect of reanalysis data – these caveats should be discussed.

Fig S6: I do not see the necessity of this figure. A small table could convey this clearly – it's simply a coin-flip probability.

L374: How much persistence is there in P-E? In other words, how large is n_2 ? How would exclusion of this term affect the result?

L414 and elsewhere: "cross term" is an unusual name – in classical Reynolds decomposition this would be called the "second order" or "nonlinear" term.

L430: Odd that the authors would not share code while the paper is in review – this seems the most critical time to do so. Will check with editorial board policies on FAIR data and code practices.

Table S2: There are DOIs for each of the CMIP6 model data sets – they should be included in the table.

Reviewer #2 (Remarks to the Author):

This is a sophisticated study on the cause of decreased seasonality (i.e. wet seasons get drier and dry seasons get wetter) over 20% land areas under future climate change. Most other regions show increased seasonality. The soil moisture-atmosphere feedbacks is demonstrated to be a main cause of the decrease seasonality in those regions. This is an interesting and noteworthy study. The paper is well-written and the methods are solid. I only have a few concerns for the authors to address.

1. It is told in the paper that SM in LFMIP-pdLC models are prescribed as the mean seasonal cycle over 1980-2014 from the historical simulation of each model, but Figure 2g-i show the soil moisture change from LFMIP-pdLC experiment. Did I miss something? And why the SM anomalies in LFMIP-pdLC are stronger than in CMIP6.

2. Figure 4 clearly shows the that increased SM can reduce P-E in dry season (denoted SM->P-E), while Figure 2 shows soil moisture change (dS). The actual effect of SM change is $dS*(SM->P-E)$ (shown in Figure 3). There lacks a discussion that soil moisture change (dS) does contribute to the P-E reduction in dry season. For example, as SM->P-E is mostly negative (Figure 4a), dS is mostly negative in RS regions (Figure 2), so $dS*(SM->P-E)$ is positive in those regions (increased water availability in dry season). This linkage and regional correspondence should be more clearly shown, probably with some new figures.

3. The paper emphasizes (in the title, abstract and text) that the reduction of seasonality is mainly caused by soil moisture feedback. This is mainly concluded according to the intensity of P-E change (Figure 3i). It seems also true for the area change of RS (Figs 3c, 3f).

Other minor comments:

Line 324 2080-2000, should be 2080-2100.

Figure S5, caption. remove "Fig.3a-c"

Figure S7, caption, remove "(d-f)"

Line 251. Should be simply "Discussion"

Reviewer #1 (Comments for the Author):

The authors analyze CMIP5 and CMIP6 simulations in terms of their changes in P-E. The core result is an association between areas of “wet get drier, dry get wetter” predominantly over subtropical land areas and land-atmosphere feedbacks. I believe major revisions would greatly improve the quality of this study, contributing to the robustness and the understanding of the results.

Response:

We thank the reviewer for the time and effort he/she put into this study. We really appreciate the reviewer’s constructive comments, which helped improve the robustness and readability of the manuscript. In the revised manuscript, we have now better shown the dry and wet seasons in the historical and future periods. We have also included a figure to show the dynamic mechanisms of how soil moisture impacts seasonal water availability through changes in the wind field.

Major comments:

L78-83: Determining dry and wet seasons based on the extremes in consecutive 3-month precipitation totals (rather than a static set of 3 months as is done here) accounts for potential shifts in seasons in a changing climate. This is a more “honest” approach, and the smaller percentage of reduced seasonality found by this method (results currently in the supplement) is indicative of this – nature does not know of calendars. I think this should be the default way to calculate the changes, and the version fixed on the historical period seasons should be supplemental. Regardless of whether the change is made, there should be more exploration of this contrast in the paper. At a minimum, I suggest two additions: (1) a figure showing how seasons have shifted, which would require a 4-panel plot of: 2 12-color (for JFM, FMA, MAM, etc.) maps of the consensus (median?) wet and dry seasons; and 2 maps of the average shift in months from historical to future; (2) As a companion to Figs 1 and S1, produce another parallel calculation based on using seasons defined based on the future period – this would close the assessment of the sensitivity of the results to the definitions of wet and dry seasons. I suspect the smallest changes will be when shifts in seasons are accounted for, rather than misdefining the seasons in one or the other set of experiments.

Response:

In this study, we re-examined the “wet seasons get wetter, dry seasons get drier” (WWDD) paradigm using CMIP5 and CMIP6 projections. While the WWDD pattern does broadly hold over much of the global land surface, we also find wet seasons become drier and dry seasons become wetter (WDDW) over subtropical regions and the Amazon, and identified the underlying mechanisms of soil moisture-atmosphere feedbacks.

To identify seasonal P-E changes between historical and future periods, we used two ways to define dry and wet seasons. We first focused on historical dry and wet seasons and compared P-E in fixed seasons between historical and future periods. We understand that the seasonal cycle may shift in the future, as suggested by the reviewer, so we also identified wet and dry seasons in the historical and future periods individually. Our original results show that ~20% of land area will experience reduced P-E seasonality by applying a historical definition to both periods (Fig. 1). The land area percentage is smaller (~10%) if we account for changes in dry and wet seasons between historical and future periods (Fig. S1 in the original supplement), but the spatial patterns of seasonal P-E changes are quite similar.

We totally agree with the reviewer that it is necessary to explore the impact of seasonal definition on the results. According to the reviewer's suggestion, we have now included a figure to show historical dry and wet seasons and their shifts in the future period (Fig. S1 in the revised supplement).

Figure S1. Shifts in dry and wet seasons in CMIP5 and CMIP6. a,b, Historical dry and wet seasons in CMIP5 based on P-E. The median month of each season is shown in color. c,d,

Future shifts in dry and wet seasons in CMIP5, with negative and positive values indicating future shifts to earlier and later months, respectively. **e-h**, The same as **a-d**, but for CMIP6.

There are two reasons that motivate our use of the historical definitions of dry and wet seasons for both periods in our primary analyses. First, fixed dry and wet seasons are required to isolate the soil moisture (SM) effect on seasonal P-E changes ($\Delta(P-E)$) between CMIP6 and LFMIP-pdLC. As SM in LFMIP-pdLC was prescribed as the mean seasonal cycle over 1980-2014 from the historical simulation, SM changes between historical and future periods are zero if we use fixed dry and wet seasons. Therefore, it is reasonable to attribute the difference in $\Delta(P-E)$ between CMIP6 and LFMIP-pdLC to the SM effect. However, if we allow dry and wet seasons to change, SM in the dry and wet seasons would change accordingly in LFMIP-pdLC, resulting in uncertainty in the identified SM effect on seasonal P-E changes. Second, we compared seasonal P-E changes between historical and future periods to examine the extent to which the WWDD paradigm holds. The WWDD paradigm is derived from a simple thermodynamic scaling of P-E, which says that future changes in P-E are proportional to changes in local near surface air temperature (ΔT) and climatological P-E in the historical period, namely $\Delta(P - E) \approx \alpha \cdot \Delta T \cdot (P - E)$, where α is a temperature-dependent parameter derived from the Clausius–Clapeyron relationship¹. This simple thermodynamic scaling indicates that historical wet seasons (P-E>0) will get wetter and historical dry seasons (P-E<0) will get drier. Following the WWDD paradigm, we first identified how P-E would change for wet and dry seasons defined according to historical simulations.

For the above reasons, we continue to use the historical definition in the main text. We also tested the seasonal changes in P-E if we use fixed dry and wet seasons according to future simulations, and find only ~5% of land area will experience reduced P-E seasonality (Fig. R1). However, this kind of definition essentially addresses the question of whether P-E would change in future dry and wet seasons but does not consider dry and wet seasons in the historical (current) period, and is thus not closely related to the WWDD paradigm. Therefore, we did not use this definition in the study.

Figure R1. Multi-model mean seasonal changes in water availability in CMIP5 and CMIP6. The same as Fig. 1 but the dry/wet season is defined as three consecutive months with lowest/highest mean P-E in the future period for each model.

Specific comments:

L70: Kumar et al (2014) is cited, but even more germane to this study is Kumar et al. (2015; [https://doi.org/ 10.1002/2015GL066858](https://doi.org/10.1002/2015GL066858)).

Response:

Both Kumar et al. (2014) and Kumar et al. (2015) have been cited now (references 11 and 14).

Much of what it is Fig S2 should be in the main manuscript, as it is crucial to explaining the results shown in Fig 3. I think the contents of Fig 4 are largely intuitive and less crucial than Fig S2 – Fig 4 could be moved to the supplement as supporting material.

Response:

Fig. S2 shows the global patterns of the SM effects on the components of P-E, including P, E, MC and the three components of MC, in the wet and dry seasons. In the main manuscript, we have included the area weighted mean seasonal changes in P, E, MC and the three components of MC in the enhanced and reduced seasonality regions in Fig. 5, which is closely related to Fig. S2 and clearly shows the most important information from Fig. S2. Therefore, we put Fig. S2 in the supplement.

In this study, we identified the SM effects on P-E using both modelling and statistical methods. Fig. 3 shows results from the modelling experiment, while Fig. 4 provides evidence from the empirical statistical perspective. The consistent results from both methods strengthen the robustness of our study. Given the importance of Fig. 4 for supporting the modelling results, we think it is better to show it in the main manuscript.

L229 & 260-261: What about a change in soil moisture leads to the changes in winds/advection? It must be though changes in atmospheric heating (changing temperatures hypsometrically affecting thickness, geopotential) via soil moisture controls on surface heating. The flow changes in the subtropics must still be largely geostrophic, and not pressure gradient as conjectured at L260. Again, this can be verified with the data in hand – a streamfunction calculation would quantify this nicely.

Response:

In the manuscript, we have investigated the mechanisms of how SM changes modify atmospheric humidity and wind fields to impact moisture convergence and P-E. The SM effect on P-E through horizontal wind changes was identified as the dynamic term of P-E changes ($-\frac{1}{\rho_w g} \nabla \cdot \int_0^{p_s} (\bar{q}_0 \cdot \Delta \bar{\mathbf{u}}) dp$) shown in Fig. 5 and Fig. S2 in the original manuscript.

To better illustrate the dynamic effect, we also show how SM changes impact negative pressure velocity ($-\omega$) throughout the troposphere (Fig. S4 in the original supplement), as the dynamic term is approximately equal to vertical moisture advection ($-\frac{1}{\rho_w g} \int_0^{p_s} (\Delta \bar{\omega} \partial q / \partial p) dp$) according to the mass continuity equation².

Following the reviewer's suggestion, we have added a supporting figure of SM-induced horizontal wind changes in the revised supplement (Fig. S4). The reviewer is correct that SM induced flow changes in the subtropics is largely geostrophic. SM induced land warming and land-ocean pressure gradient (Fig. S5) further promote flow convergence and vertical ascent in the lower and middle troposphere over many subtropical dry regions (Figs. S4 and S6).

Figure S4. SM effects on tropospheric winds in the wet and dry seasons in CMIP6. a-d, The SM effect on the change in horizontal winds (CMIP6 minus LFMIP-pdLC) between 1980-2000 and 2080-2100 in the wet season. **e-h,** The same as **a-d**, but for the dry season.

We have also explained the dynamic effect in the revised manuscript.

“As the dynamic term $(-\frac{1}{\rho_w g} \nabla \cdot \int_0^{p_s} (\bar{q}_0 \cdot \Delta \bar{u}) dp)$ is approximately equal to $-\frac{1}{\rho_w g} \int_0^{p_s} (\Delta \bar{\omega} \partial q / \partial p) dp$, where ω is the pressure vertical velocity, according to the mass continuity equation⁵³, we identify the dynamic effect by analyzing the SM effects on both the horizontal wind velocity and the pressure vertical velocity throughout the troposphere.”

Fig 5: Panels g and h: I did not catch at first the minus sign in front of omega until I had looked at the figure several times. It would be helpful to draw vertical arrows next to the pressure levels that show the biggest departure from 0 in the RS cases, just to make it very clear which is associated with ascent and descent.

Response:

In Fig. 5, positive and negative values of the negative pressure velocity ($-\omega$) are associated with vertical ascent and descent, respectively. We have added vertical arrows to show vertical motions in Fig. 5g,h and included description of the arrows in figure caption. This does make the figure easier to interpret. Thank you for this suggestion.

Figure 5. Mechanisms of the SM effect on water availability in the wet and dry seasons. **a-f**, SM effects on changes in precipitation, evapotranspiration, moisture convergence, and the thermodynamic, dynamic, and nonlinear terms of moisture convergence between 1980-2000 and 2080-2100 in enhanced seasonality (ES) regions and reduced seasonality (RS) regions shown in Fig. 3c. The SM effect is assessed as the five-model mean difference between CMIP6 and LFMIP-pdLC (similar to Fig. 3g-i). **g,h**, SM effects on negative pressure velocity ($-\Delta\omega$) over ES and RS regions. The error bars in **a-f** and the shading in **g** and **h** show the standard

deviation of each variable across the five models. The upward and downward arrows in **g** and **h** show vertical ascent and descent, respectively.

L258: The speculation about recycling can be checked. It would be straightforward to follow one of the methodologies of bulk moisture transport (e.g., Brubaker et al. 1993; [https://doi.org/10.1175/1520-0442\(1993\)006<1077:EOCPR>2.0.CO;2](https://doi.org/10.1175/1520-0442(1993)006<1077:EOCPR>2.0.CO;2), or alternatively Trenberth 1999; [https://doi.org/https://doi.org/10.1175/1520-0442\(1999\)012<1368:AMRROA>2.0.CO;2](https://doi.org/https://doi.org/10.1175/1520-0442(1999)012<1368:AMRROA>2.0.CO;2)) which are appropriate to monthly model output.

Response:

We are hoping to look at these regional recycling issues in greater detail in the future using the Water Accounting Model³ (van der Ent et al., 2014) that was used in our previous study⁴ (Findell et al. 2019) for continental-scale investigations. In this study, we focused on the dynamic and thermodynamic mechanisms of the SM effect on moisture convergence and P-E. We summarized the main finding of a dominant dynamic mechanism of seasonally varying SM effects on P-E in the discussion. Our discussion of the evaporated moisture recycling for precipitation arose from previous studies on precipitation recycling, which have been extensively investigated and well recognized⁵⁻⁹. We have now cited two related references^{5,6} of SM effects on evapotranspiration and precipitation recycling to support the statement in the discussion.

“The resultant drying of soil in turn reduces evapotranspiration and recycling of evaporated moisture for subsequent precipitation^{24,25}.”

L283-290: The results also imply that irrigation in dry areas would compound aridity – something already seen observationally in places like the upper Ganges and Indus valleys. While speculating on the broader impacts of your results, this could be mentioned.

Response:

We have included an additional discussion on irrigation in the last paragraph of the Discussion section.

“As the soil moisture feedback on surface water availability is stronger in drier conditions, human activities like irrigation may weaken the negative soil moisture feedback and magnify water scarcity in dry regions, but the scale at which this effect might be detected need to be identified.”

Regarding use of reanalysis datasets: data assimilation violates conservation and budget closure (not a problem with the free-running CMIP models). This affects the divergent part of

flow and transport more than the rotational part, which means moisture flux divergence is less reliable in reanalysis budgets than transport. This is also why reanalysis precipitation biases remain poor, as they are strongly connected to moisture flux convergence. Furthermore, surface fluxes are completely unconstrained by observations. So be wary of putting too much trust in this aspect of reanalysis data – these caveats should be discussed.

Response:

We agree with the reviewer that the data assimilation system may lead to violations of mass/energy conservation, resulting in uncertainties in the simulated trends of hydrological variables. To reduce the impact of these issues, we have removed the linear trends and seasonal cycles of SM and P-E to identify the SM-(P-E) feedback in the original manuscript. We note some degree of uncertainty may still exist in reanalysis datasets and CMIP6 models. As shown in Fig. 4, the strength of SM-atmosphere feedbacks vary across reanalysis products and models, which may arise from model uncertainties of soil moisture-atmosphere feedbacks, although the sign is largely consistent. We have included some discussion on the uncertainty in the revised manuscript.

“While the negative sign of the SM-(P-E) feedback is supported by both reanalysis products and CMIP6 models, the feedback strength and the magnitude of SM induced P-E changes vary across models and products (Figs. 3 and 4), which may result from uncertainties in the representation of SM-atmosphere feedbacks.”

Fig S6: I do not see the necessity of this figure. A small table could convey this clearly – it’s simply a coin-flip probability.

Response:

We agree with the reviewer and have removed Fig. S6 from the supplementary materials. To better explain the significance test, we have now included two equations (2 and 3) in the revised Methods section.

$$P_{CMIP5}(m \geq 23) = \sum_{m=23}^{35} \frac{35!}{m!(35-m)!} 0.5^m \times 0.5^{35-m} < 0.05 \quad (2)$$

$$P_{CMIP6}(m \geq 20) = \sum_{m=20}^{30} \frac{30!}{m!(30-m)!} 0.5^m \times 0.5^{30-m} < 0.05 \quad (3)$$

L374: How much persistence is there in P-E? In other words, how large is n_2 ? How would exclusion of this term affect the result?

Response:

To test the influence of P-E persistence on the SM-(P-E) feedback, we compared the sensitivity coefficient for SM→(P-E) obtained from the original regression model in Eq. (2) in the original manuscript and from a linear model that does not remove the effect of P-E persistence in Eq. (R1) below.

$$(P - E)_d(t) = n_0 + n_1 \cdot SM_d(t - 1) \quad (R1)$$

Although both models identify negative sensitivity coefficients for SM→(P-E) in the dry season, the strength of the negative feedback would be reduced if we do not remove the effect of P-E persistence.

Figure R2. Soil moisture feedbacks on water availability in the wet and dry seasons. a-b, Mean sensitivity coefficients for SM→(P-E) identified from reanalysis products ERA5 (1979-2019) and MERRA-2 (1980-2019) (the same as Fig. 4g,j). **c-d,** The same as a-b, but do not remove the effect of P-E persistence in the regression model.

L414 and elsewhere: “cross term” is an unusual name – in classical Reynolds decomposition this would be called the “second order” or “nonlinear” term.

Response:

We have replaced “cross term” with “nonlinear term” in the main text, methods section, and figure captions.

L430: Odd that the authors would not share code while the paper is in review – this seems the most critical time to do so. Will check with editorial board policies on FAIR data and code practices.

Response:

We have publicly shared our code (<https://doi.org/10.5281/zenodo.6301777>) and revised the code availability statement.

“**Code availability.** The R code used for modelling and reanalysis data analyses is publicly available (<https://doi.org/10.5281/zenodo.6301777>).”

Table S2: There are DOIs for each of the CMIP6 model data sets – they should be included in the table.

Response:

We have now included DOIs for the CMIP6 data sets in Table S2 of the revised manuscript.

Reviewer #2 (Remarks to the Author):

This is a sophisticated study on the cause of decreased seasonality (i.e. wet seasons get drier and dry seasons get wetter) over 20% land areas under future climate change. Most other regions show increased seasonality. The soil moisture-atmosphere feedbacks is demonstrated to be a main cause of the decrease seasonality in those regions. This is an interesting and noteworthy study. The paper is well-written and the methods are solid. I only have a few concerns for the authors to address.

Response:

We appreciate the reviewer’s positive assessment and valuable suggestions for improving our manuscript.

1. It is told in the paper that SM in LFMIP-pdLC models are prescribed as the mean seasonal cycle over 1980-2014 from the historical simulation of each model, but Figure 2g-i show the soil moisture change from LFMIP-pdLC experiment. Did I miss something? And why the SM anomalies in LFMIP-pdLC are stronger than in CMIP6.

Response:

It is true that SM in LFMIP-pdLC was prescribed as the mean seasonal cycle of 1980-2014 from the historical simulation in each model. In Fig. 2g-i, we show SM changes between historical (1980-2000) and SSP585 (2080-2100) simulations for the five CMIP6 models that participated in the LFMIP-pdLC experiment, rather than SM changes in the LFMIP-pdLC experiment. Fig. 2 shows stronger SM anomalies in the 5-model ensemble (Fig. 2g-i) compared to the 30-model ensemble (Fig. 2d-f). We realize that our original figure caption was unclear and have revised it as follows:

Figure 2. Multi-model mean seasonal changes in soil moisture in CMIP5 and CMIP6. The percent change in soil moisture (ΔSM) is assessed between 1971-2000 (historical simulation) and 2071-2100 (RCP8.5 or SSP585 simulation) in 35 CMIP5 models (a-c) and 30 CMIP6 models (d-f), and between 1980-2000 (historical simulation) and 2080-2100 (SSP585 simulation) in the five CMIP6 models which participated in the LFMIP-pdLC experiment (g-i). SM in the LFMIP-pdLC experiment is prescribed as the mean seasonal cycle of 1980-2014 for both historical and future periods.

2. Figure 4 clearly shows that increased SM can reduce P-E in dry season (denoted SM->P-E), while Figure 2 shows soil moisture change (dS). The actual effect of SM change is $dS*(SM->P-E)$ (shown in Figure 3). There lacks a discussion that soil moisture change (dS) does contribute to the P-E reduction in dry season. For example, as SM->P-E is mostly negative (Figure 4a), dS is mostly negative in RS regions (Figure 2), so $dS*(SM->P-E)$ is positive in those regions (increased water availability in dry season). This linkage and regional correspondence should be more clearly shown, probably with some new figures.

Response:

We agree that clarifying the linkage between the modelling and empirical statistical approaches used in our study is a good idea. In Fig. 4 and the associated discussion, we mainly compared the SM-atmosphere feedbacks between dry and wet seasons and show a consistently stronger SM effect on P-E in the dry season than in the wet season. Following the reviewer's suggestion, we have added discussion on the consistency of the modelling and statistical approaches in the strong SM effect on dry season P-E changes over the subtropics and the Amazon.

“While the SM effect on long-term P-E changes in subtropical dry regions has been recognized²², our modelling and empirical assessments further show that the negative SM-(P-E) feedback occurs mainly in the dry season. The projected SM declines combined with the negative SM-(P-E) feedback well explains the positive SM effect on dry season P-E increases isolated using modelling experiments over subtropical regions and the Amazon.”

3. The paper emphasizes (in the title, abstract and text) that the reduction of seasonality is mainly caused by soil moisture feedback. This is mainly concluded according to the intensity of P-E change (Figure 3i). It seems also true for the area change of RS (Figs 3c, 3f).

Response:

The reviewer is correct. In the five CMIP6 coupled climate models, 18% of land area shows reduced seasonality of P-E (Fig. 3c), while in LFMIP-pdLC without SM feedbacks, the percentage of reduced seasonality regions is reduced to 9% (Fig. 3f). We have now included the results in the revised manuscript.

“Although the magnitude of SM drying is similar in both seasons (Fig. 2g,h), the negative SM feedback on P-E is much weaker in the wet season than the dry season (Fig. 3g,h), contributing to the WDDW pattern and reduced seasonality of P-E over 18% of land area in CMIP6 coupled simulations (Fig. 3c). Without the SM effect in LFMIP-pdLC, only 9% of land area experiences reduced seasonality of P-E (Fig. 3f).”

Other minor comments:

Line 324 2080-2000, should be 2080-2100.

Response:

We have corrected the study periods accordingly.

“Although long-term hydrological changes are usually assessed at 30-year time scales (Fig. 1), we compared seasonal changes in precipitation, evapotranspiration, and P-E between 1971-2000 and 2071-2100 and between 1980-2000 and 2080-2100 in CMIP6, and found the results are very close (Fig. S7).”

Figure S5, caption. remove “Fig.3a-c”

Response:

The figure caption has been revised accordingly.

Figure S7, caption, remove “(d-f)”

Response:

The figure caption has been revised accordingly.

Line 251. Should be simply “Discussion”

Response:

The section title has been revised accordingly.

References:

1. Held, I. M. & Soden, B. J. Robust Responses of the Hydrological Cycle to Global Warming. *J. Climate* **19**, 5686–5699 (2006).
2. Bony, S. *et al.* Robust direct effect of carbon dioxide on tropical circulation and regional precipitation. *Nature Geoscience* **6**, 447–451 (2013).
3. van der Ent, R. J., Wang-Erlandsson, L., Keys, P. W. & Savenije, H. H. G. Contrasting roles of interception and transpiration in the hydrological cycle – Part 2: Moisture recycling. *Earth Syst. Dynam.* **5**, 471–489 (2014).
4. Findell, K. L. *et al.* Rising Temperatures Increase Importance of Oceanic Evaporation as a Source for Continental Precipitation. *Journal of Climate* **32**, 7713–7726 (2019).
5. Seneviratne, S. I. *et al.* Investigating soil moisture–climate interactions in a changing climate: A review. *Earth-Science Reviews* **99**, 125–161 (2010).
6. Dirmeyer, P. A., Schlosser, C. A. & Brubaker, K. L. Precipitation, Recycling, and Land Memory: An Integrated Analysis. *J. Hydrometeor.* **10**, 278–288 (2009).
7. Ent, R. J. van der & Savenije, H. H. G. Length and time scales of atmospheric moisture recycling. *Atmospheric Chemistry and Physics* **11**, 1853–1863 (2011).

8. Gimeno, L. *et al.* Recent progress on the sources of continental precipitation as revealed by moisture transport analysis. *Earth-Science Reviews* **201**, 103070 (2020).
9. Trenberth, K. E. Atmospheric Moisture Recycling: Role of Advection and Local Evaporation. *J. Climate* **12**, 1368–1381 (1999).

REVIEWER COMMENTS

Reviewer #2 (Remarks to the Author):

In the LFMIP-pdLC experiments, soil moisture (SM) was prescribed as the mean seasonal cycle of 1980-2014 from the historical simulation in each model. Then the seasonal changes in P-E and other variables between 1980-2000 and 2080-2100 were compared. The difference between CMIP6 (includes SM feedback) and LFMIP-pdLC in P-E season cycle is argued to the effect of SM feedback. However, as SM in 1980-2014 is prescribed for the whole 1980-2100 period, the SM effect also includes a change in SM climatology. The LFMIP-pdLC experiment cannot separate this effect. Another LFMIP experiment, LFMIP-rmLC, prescribes 30-year monthly running means from the historical simulation to each model. The difference between CMIP6 and LFMIP-rmLC is the effect of SM feedback without the effect of SM climatology change. Hope the authors can address this issue.

Reviewer #3 (Remarks to the Author):

This study carefully picks apart and analyzes the surprising result that over large areas of the subtropics and Amazon basin, global warming tends to make the dry season wetter and the wet season drier (in the sense of precipitation-minus-evaporation) in multi-climate model projections. It finds a key role for the suppression of dry-season evaporation by drying soil moisture (SM). The analysis is carried out expertly, and I learned a lot from reading it.

However, I have an alternative interpretation/framing of the study's results that I think is a lot more complete, simpler to understand, and may contain more insight (immediately below). Thus, I recommend major revisions so that this interpretation can be incorporated in addition to (or instead of) the authors' first interpretation. There was also a major accidental error with one of the figures in the supplement, which rendered that figure almost unreadable and needs to be fixed.

Finally, I have a few minor writing and graphical suggestions that should also be addressed, at the bottom.

I should add that I am a *new* reviewer who did not review the initial submission. I was added to replace a reviewer from the first round who didn't have time to review the second round. Thus, it's my first time seeing the manuscript, and this review should be thought of as an initial-submission review (not a review of revision).

Major/Framing issue:

32-35, 175-177, 262-264, 270-274 etc: This is indeed one way to look at it - but one could just as easily break down the annual-range reduction by *season* first, rather than by SM vs non-SM. When it's framed that way, the wet season P-E reductions are clearly more important than the dry season P-E increases! This can clearly be seen in Fig 3j-l's RS Regions red bars: the wet season red downward bar (3j) is quite large and is nearly equal to the annual-range red downward bar (3l), while the dry season red upward bar (3k) is small. (This is also pretty clear just looking at the intensity of the color in the WDDW regions in Fig 1a,b vs Fig 1c,d). In turn, 3j shows that the wet season red downward bar is dominated by non-SM effects (the adjacent yellow downward bar). So in this latter framing, the most important process by far is non-SM-driven P-E reductions during the wet season. In this way, paradoxically, we can get two very different answers to your question depending on whether we break by season first (wet season non-SM effect seems most important) or whether we break by process first, as you did here (dry season SM effect seems most important). This "paradox" is possible because we have multiple terms of different signs adding up and being split different

ways.

Therefore, to avoid this ambiguity, I think the only fair way to answer your question is to think of the total RS Regions seasonality reduction (i.e. red downward bar in 3l) as the sum of *four independent* contributions: wet season non-SM effect (big yellow downward bar in 3j), wet season SM effect (small blue downward bar in 3j), dry season non-SM effect (inverse of small yellow downward bar in 3k), and dry season SM effect (inverse of big blue upward bar in 3k). By definition, these four add to the total seasonality reduction. But now all four can be compared fairly and independently. Focusing on those four bars, we see that wet season non-SM effect (3j RS yellow) is the largest, reducing seasonality about -0.4 mm/day. But dry season SM effect (3k RS blue) is almost as large, reducing seasonality about -0.3 mm/day (I flipped the sign to emphasize seasonality reduction). The remaining two bars are much smaller, and their effects on seasonality largely cancel out.

So, if I were you, I would frame your key takeaway not as "dry season SM effect dominates the seasonality reduction", but rather as, "wet season non-SM effect and dry season SM effect together dominate the seasonality reduction, with the former slightly more important". I think that is a lot fairer.

Now, to speculate a bit, what might be driving these two key effects?

For the wet season non-SM effect on P-E (3j RS downward yellow bar), I would imagine it is mostly just P reduction. That's because with SM fixed by the pdLC protocol, E won't be able to change too much with climate change (while we know that in most of the RS regions, P is indeed projected to strongly decline in the CMIP average). But you could check this speculation of mine easily.

For the dry season SM effect on P-E (3k RS upward blue bar), Fig S3b,e shows that it's dominated by E reduction, not P increase (as you again show quite clearly in Figure 4). So the SM-circulation-P feedback mechanism you invoke at e.g. 120-121 doesn't seem to be that important. Instead, it seems to be a much more straightforward story: lower annual and/or wet-season P leads to lower dry-season SM, which curtails dry-season E. Even more fundamentally, if P lowers a lot, E must also lower a lot just by water mass conservation, if we are in a water-limited region, which many of these places are. This will show up in the SM effect, rather than in the non-SM effect.

So I think the regional P-E seasonality reduction is happening for one very simple ultimate reason - *reduced total and/or wet-season P*. This both reduces wet-season P-E, and also increases dry-season P-E (makes it "less negative") by reducing the supply of water mass (soil moisture) available to supply dry-season E, while dry-season P is ~ 0 and can't change much.

If this does not make sense, think of some simple end-member examples. In a climate with no P (e.g. Saharan interior), there will be no E either. So the seasonality of P-E will be zero. On the other hand, in a climate with moderate amount of P that falls seasonally (e.g. Sahel), P-E seasonality will be decently large because wet season P-E will be positive while dry season P-E will be negative (as plants transpire some of the water that fell in the wet season, even though no water is currently falling). Finally in a climate with very high P that falls seasonally (e.g. northeastern India or Bangladesh, or parts of subtropical South America), P-E seasonality will be gigantic, because wet season P-E will be very positive but dry season P-E will be quite negative due to the combination of \sim zero P with "wet" E from the saturated soil and groundwater. So in climates with highly seasonal P (i.e. most of your WDDW regions), the wet-season or annual P magnitude should be a very strong control on the P-E seasonality, largely via its effects on wet-season P and on dry-season E.

You do not have to completely adopt this framing, but I think you at least must mention and discuss it substantially. It would be a much simpler explanation for your results

than the explanation you are giving!

Similarly, I would interpret the result at 223-224 not as increased moisture convergence, but rather as **decreased** moisture **divergence** stemming simply from the lower E. In the dry season, the key balance is that E comes out of the land surface and is diverged away by the wind (little of it is recycled as P due to lack of upward motion in dry season). So if E declines, then the moisture divergence will also have to decline. Again it is a simple mass conservation argument.

It is very interesting that this dry season moisture divergence decline is accomplished by the SH-driven decline in the strength of the subsidence (blue curve in Fig 5h; purple bar in Fig 5d) rather than a decline in the thermodynamic term, but still I think the water mass conservation has to hold. Presumably, the profile of dry season -w itself (as opposed to -deltaw) is sinking/subsidence both in the present and future climates, it's just weaker subsidence in the future (which would explain why P is almost unchanged, rather than increasing from the increase in -w as one might expect).

Finally, in this view, the assessment at 288-289 and 300 may be too optimistic. If what's really happening is just reduction in dry-season E due to lack of water supply, then that's clearly a "droughty" trend, not an anti-droughty trend. I.e. it would force plant productivity **declines**, not increases, seeing that dry-season E mainly represents plant water use for carbon assimilation. (And would also lead to runoff declines, since the driver is the lack of water supply [SM] in the first place, which would cause **both** E and Q to decline in the dry season - again you could easily check this using model runoff fields).

Major plotting error:

Fig S6a-h color bar's interior is accidentally transposed in x-y space -- the colors vary **vertically** within the colorbar, instead of horizontally. This makes it impossible to figure out whether red=positive or red=negative. I inferred that red=negative, but I am not sure! This must be fixed to be readable.

Minor:

44: Ref 5 quite clearly used the phrase "intensification of the global water cycle" to mean the 1-2% increase in **global-mean P** or **global-mean E**, **not** the much more important 7% amplification of spatial/seasonal P-E pattern that they derived later in their paper and that you're discussing here. Some writers still follow this usage from Ref 5, and some do not, leading to confusion. Therefore, for maximum clarity to all readers, I would just delete this phrase, ending up with e.g. "Global warming increases water vapor in the atmosphere. This increase is generally expected to amplify the existing..." and then cite 5,6,7,8 at the end of the second sentence.

69: "promotes" is a bit of an odd word choice; something like "amplifies" or "intensifies" would be more usual.

71: similarly "reflect" is odd here - what about just "simulate" or "project"?

81: for maximum clarity, should be "smaller percentage (~10 percentage points) of reduced seasonality regions", to make it clear that you mean e.g. 30% reducing to 20%, not 30% reducing to 27%.

104-106: should emphasize that this is not true in general over land (e.g. Byrne and O'Gorman 2015, Greve and Seneviratne 2015).

Fig 2: it would be very useful to outline the WDDW regions on the panels (e.g. with a

thin black contour), so as to make it easy to quickly verify the statement at 127-129.

Fig 2: It makes no sense to plot % change in a quantity like SM that can be ~zero (that's why you have huge signals over e.g. the Sahara Desert that dominate the map). A tiny and irrelevant change in a near-zero quantity could be a huge % change. Rather, you should plot standardized change (i.e. SM change normalized by 1971-2000 SM interannual standard deviation) or dimensionless/volumetric change (i.e. change in SM divided by the depth of the soil layer over which SM is quantified). Those would each be much less prone to spurious large values in deserts than % change.

end 141-143: This last sentence made me quite confused for a while, because I thought, if SM is prescribed to be the same thing in both the historical and the future period, then how could there be any SM difference between the two periods at all? I.e. why isn't the last row of Figure 2 solid white, given this? It took me some time to realize that the last row of Figure 2 is actually the results from the *regular*, *standard* CMIP6 historical and SSP585 simulations performed by the LFMIP-pdLC-participating models (*not* the LFMIP-pdLC simulations themselves). So you must make that explicit.

159: should be (Fig. S3d,e). The reader should not have to hunt for the specific panels you're referring to.

161-162: With the important exceptions of the Amazon and Australia - these should be noted.

Fig 5g,h: Red and blue colors should not be used for the arrows, since they are already used to denote ES and RS respectively. Rather, the arrows should be black so as not to confuse the viewer. The direction of the arrow alone is sufficient to explain its meaning; color (which could be confused with your ES and RS colors) isn't needed.

236: I don't quite see what Fig S4 is supposed to be showing, or how to draw the conclusion here from Fig S4? You may have to walk the reader through this more. It's much easier to see in Fig. 5b,d,f.

238: should be (Fig. S5b,d,f).

319-320: is there a reason that E was calculated from LH rather than taken from the CMIP output directly? The CMIP output does usually include E, as "evspsbl". The reason you did not use "evspsbl" should be explained in the text. If there's no reason, then you should directly use "evspsbl" rather than calculating from LH.

390-401: The use of lag correlation here seems odd - the E (and thus P-E) should depend on the current SM, not the SM of one month ago. Did you test to see if the correlations became stronger if zero-lag was used instead of 1-month-lag? This should be mentioned in the text as well.

Fig S1, top row: The regions with December- and January-centered wet seasons are a bit hard to read on these maps because the plotting function attempts to squish all the other months in between the January and December regions, leading to shimmering, stringy rainbow artifacts at all the December-January (purple-to-blue) boundaries. You can especially see this in e.g. the Middle East, Central Asia, and the Canada-US border region in Fig S1e. So is there a way to use a less-fancy plotting function that does not try to interpolate or draw contours, but instead just plots the discrete value at each pixel? That would make the December-January boundary much sharper and looking more like the remaining months' boundaries. (In matlab you would just use pcolor or imagesc instead of contourf, but I don't know what environment you are using here). Note that this suggestion is optional, but would help a lot if doable.

To be complete, this suggestion also applies to the second row, but for whatever reason the problem is more visually apparent in the first row (the wet season map).

Fig S1, third and fourth rows: The color scale here could really benefit from a couple of improvements. First, the value 0 should be given a completely neutral color (e.g. white), not yellow - since yellow fools the viewer into thinking it's a negative shift. Second, because shifts of -5 and of +6 mean almost the same thing in practice, it makes no sense to designate them with opposite colors. Rather, I would assign +6 a shade of intense purple (equidistant between red and blue), and then move the intense red color from -5 to -3 (and the intense blue color from +6 to +3). Finally I would interpolate between these to get the remaining months.

In short: 0=white -> -3=red -> 6=purple -> +3=blue -> 0=white, with the arrows representing interpolation in the intervening months. It may work even better if the red and blue are assigned to -4 and +4 respectively, you could try that as well (0=white -> -4=red -> 6=purple -> +4=blue -> 0=white).

Reviewer #2 (Comments for the Author):

In the LFMIP-pdLC experiments, soil moisture (SM) was prescribed as the mean seasonal cycle of 1980-2014 from the historical simulation in each model. Then the seasonal changes in P-E and other variables between 1980-2000 and 2080-2100 were compared. The difference between CMIP6 (includes SM feedback) and LFMIP-pdLC in P-E season cycle is argued to the effect of SM feedback. However, as SM in 1980-2014 is prescribed for the whole 1980-2100 period, the SM effect also includes a change in SM climatology. The LFMIP-pdLC experiment cannot separate this effect. Another LFMIP experiment, LFMIP-rmLC, prescribes 30-year monthly running means from the historical simulation to each model. The difference between CMIP6 and LFMIP-rmLC is the effect of SM feedback without the effect of SM climatology change. Hope the authors can address this issue.

Response:

We thank the reviewer for the constructive comment. In this study, we isolated the SM effects on P-E changes ($\Delta(P-E)$, future minus historical P-E) in the dry and wet seasons by comparing CMIP6 and LFMIP-pdLC. In LFMIP-pdLC, SM was prescribed as the mean seasonal cycle of 1980-2014 from the historical simulation, and SM changes between historical and future periods are zero for both dry and wet seasons. Therefore, we can attribute the difference in $\Delta(P-E)$ between CMIP6 and LFMIP-pdLC to the SM effect (changes in the mean SM and SM variability) in the dry and wet seasons, respectively.

In LFMIP-rmLC, SM was prescribed as 30-year monthly running means of SM from historical and SSP585 simulations, while the monthly and interannual variability of SM was removed. The difference in $\Delta(P-E)$ between CMIP6 and LFMIP-rmLC therefore represents the influence of changes in SM variability, which has been shown to be very small compared to the effect of SM drying trend (changes in the mean SM) in our previous study using the GLACE-CMIP5 experiment¹, the predecessor of the LFMIP experiment in CMIP6. Therefore, we did not use the LFMIP-rmLC experiment in this study.

We agree with the reviewer that the isolated SM effect on the annual range of P-E changes (wet season $\Delta(P-E)$ minus dry season $\Delta(P-E)$) between CMIP6 and LFMIP-pdLC includes 1) changes in SM climatology (or seasonally asymmetric SM changes), 2) seasonally varying SM-(P-E) feedbacks. After careful thought, we found the LFMIP-rmLC experiment cannot be used to isolate the contribution of changes in SM climatology from that of seasonally varying SM feedbacks. Although the difference between CMIP6 and LFMIP-rmLC removes the effect of SM climatology, it also removes the effect of changes in the mean SM (dSM) in both dry and wet seasons. Therefore, the effect of seasonally vary SM-(P-E) feedbacks ($dSM \cdot \left(\frac{\partial(P-E)}{\partial SM}\right)_{wet} - dSM \cdot \left(\frac{\partial(P-E)}{\partial SM}\right)_{dry}$) cannot be quantified using the LFMIP-rmLC experiment.

In this study, we examined the effect of changes in SM climatology by comparing SM changes in the dry and wet seasons (Fig. 2), and directly compared the SM-(P-E) feedbacks between dry and wet seasons using an empirical statistical method (Fig. 4). Fig. 2 shows that SM reduces in the RS regions for both wet and dry seasons, with small inter-seasonal differences, which indicates that the distinct SM effects on P-E between wet and dry seasons are unlikely induced by changes in SM climatology. On the other hand, the results from the empirical statistical analysis in Fig.4 are highly consistent with the total SM effects identified by comparing CMIP6 and LFMIP-pdLC. We therefore believe that the distinct SM effects on P-E changes between wet and dry seasons are mainly caused by seasonally varying SM-(P-E) feedbacks. Although the effect of changes in SM climatology may be small compared to the seasonally varying SM-(P-E) feedbacks, we have pointed it out in the revised manuscript.

“The above SM effect on seasonal P-E changes highlights the seasonally varying nature of SM-(P-E) feedbacks, but also includes the effect of changes in SM climatology (i.e., mean seasonal cycle of SM, Fig. 2g-i). To directly compare the SM-(P-E) feedbacks between dry and wet seasons, we apply an empirical statistical method (Methods) to CMIP6 models and observationally constrained reanalysis products (Modern-Era Retrospective analysis for Research and Applications (MERRA-2)²⁸ and European Centre for Medium-Range Weather Forecasts (ERA5)). The five CMIP6 models and two reanalysis products consistently show strong negative SM-(P-E) feedbacks over subtropical regions in the dry season, when the positive SM effect on evapotranspiration exceeds that on precipitation (Fig. 4a-c,g-i).”

Reviewer #3 (Remarks to the Author):

This study carefully picks apart and analyzes the surprising result that over large areas of the subtropics and Amazon basin, global warming tends to make the dry season wetter and the wet season drier (in the sense of precipitation-minus-evaporation) in multi-climate model projections. It finds a key role for the suppression of dry-season evaporation by drying soil moisture (SM). The analysis is carried out expertly, and I learned a lot from reading it.

However, I have an alternative interpretation/framing of the study's results that I think is a lot more complete, simpler to understand, and may contain more insight (immediately below). Thus, I recommend major revisions so that this interpretation can be incorporated in addition to (or instead of) the authors' first interpretation. There was also a major accidental error with one of the figures in the supplement, which rendered that figure almost unreadable and needs to be fixed.

Finally, I have a few minor writing and graphical suggestions that should also be addressed, at the bottom.

I should add that I am a **new** reviewer who did not review the initial submission. I was added to replace a reviewer from the first round who didn't have time to review the second round. Thus, it's my first time seeing the manuscript, and this review should be thought of as an initial-submission review (not a review of revision).

Response:

We thank the reviewer for the time and effort he/she put into this study. We really appreciate the reviewer's thoughtful insights into the interpretation of the WDDW pattern, which helped improve the quality and readability of the manuscript. In the revised manuscript, we have incorporated the reviewer's interpretation based on water mass conservation over land as a complement to our interpretation from the perspective of land-atmosphere coupling. We have also corrected the errors in several figures in the manuscript and supporting materials.

Major/Framing issue:

32-35, 175-177, 262-264, 270-274 etc: This is indeed one way to look at it - but one could just as easily break down the annual-range reduction by **season** first, rather than by SM vs non-SM. When it's framed that way, the wet season P-E reductions are clearly more important than the dry season P-E increases! This can clearly be seen in Fig 3j-l's RS Regions red bars: the wet season red downward bar (3j) is quite large and is nearly equal to the annual-range red downward bar (3l), while the dry season red upward bar (3k) is small. (This is also pretty clear just looking at the intensity of the color in the WDDW regions in Fig 1a,b vs Fig 1c,d). In turn, 3j shows that the wet season red downward bar is dominated by non-SM effects (the adjacent yellow downward bar). So in this latter framing, the most important process by far is non-SM-driven P-E reductions during the wet season. In this way, paradoxically, we can get two very different answers to your question depending on whether we break by season first (wet season non-SM effect seems most important) or whether we break by process first, as you did here (dry season SM effect seems most important). This "paradox" is possible because we have multiple terms of different signs adding up and being split different ways.

Therefore, to avoid this ambiguity, I think the only fair way to answer your question is to think of the total RS Regions seasonality reduction (i.e. red downward bar in 3l) as the sum of **four** **independent** contributions: wet season non-SM effect (big yellow downward bar in 3j), wet season SM effect (small blue downward bar in 3j), dry season non-SM effect (inverse of small yellow downward bar in 3k), and dry season SM effect (inverse of big blue upward bar in 3k). By definition, these four add to the total seasonality reduction. But now all four can be compared fairly and independently. Focusing on those four bars, we see that wet season non-

SM effect (3j RS yellow) is the largest, reducing seasonality about -0.4 mm/day. But dry season SM effect (3k RS blue) is almost as large, reducing seasonality about -0.3 mm/day (I flipped the sign to emphasize seasonality reduction). The remaining two bars are much smaller, and their effects on seasonality largely cancel out.

So, if I were you, I would frame your key takeaway not as "dry season SM effect dominates the seasonality reduction", but rather as, "wet season non-SM effect and dry season SM effect together dominate the seasonality reduction, with the former slightly more important". I think that is a lot fairer.

Response:

In this study, we used CMIP5 and CMIP6 projections to show that wet seasons become drier and dry seasons become wetter (WDDW), from the perspective of P-E, over subtropical regions and the Amazon, and identified the underlying mechanisms of SM-atmosphere feedbacks.

We isolated the SM effect from the non-SM effect using CMIP6 and LFMIP-pdLC simulations. **The main finding of this study is that the seasonally varying SM effects on P-E, rather than dry season SM effect, dominate the reduced seasonality of P-E over subtropical regions and the Amazon.** In the reduced seasonality (RS) regions, the non-SM effect reduces P-E and SM in both seasons (Fig. 3d,e and Fig. 2g,h). SM drying strongly increases dry season P-E and cancels out P-E reductions induced by the non-SM effect (Fig. 3e,h,k); however, the negative SM feedback on P-E is much weaker in the wet season, and wet season P-E reductions are mainly caused by the non-SM effect in most RS regions (Fig. 3d,g,j). We emphasize the importance of the seasonally varying SM effects because the SM reductions are similar in magnitude between dry and wet seasons (Fig. 2g,h,i), and the SM effect on the annual range of P-E would be small if the SM feedbacks on P-E do not change seasonally. Indeed, as the non-SM effect reduces P-E in both dry and wet seasons, only 9% of land area experiences reduced seasonality of P-E in LFMIP-pdLC (Fig. 3f); however, the seasonally varying SM effects reduce the annual range of P-E over 36% of land area (Fig. 3i). Overall, in the RS regions shown in Fig. 3c, the reduced annual range of P-E is dominated by the contribution of distinct SM effects between wet and dry seasons (-0.39 mm/day), which is 63% larger than the non-SM effect (-0.24 mm/day) (Fig. 3l).

The reviewer proposed an alternative framing, i.e., decomposing P-E changes into four components, to explain the results and emphasized the importance of wet season non-SM effect and dry season SM effect from a quantitative perspective. It should be noted that the wet season non-SM effect on P-E is large in Latin America but relatively small in many other RS regions, compared to the wet season SM effect (e.g., Australia) and the dry season non-SM effect (e.g., Northern Hemisphere mid-latitude regions). Although the area-weighted mean results (Fig. 3j-

l) show that the wet season non-SM effect is the largest contributor, mainly due to the strong effect from the Latin America, it cannot well explain the reduced annual range of P-E for other RS regions.

Overall, our mechanistic understanding of the seasonally varying SM effects on P-E can better explain the reduced annual range of P-E for the RS regions. We agree with the reviewer that the wet season non-SM effect is important for the drying of the wet season and the dry season SM effect leads to the wetting of the dry season. In the revised manuscript, we have adopted the reviewer’s framework and identify the main drivers of the WDDW pattern for each season first before comparing the contributions of the SM and non-SM effects to the reduced seasonality of P-E. We have also included a new figure (Fig. S4) to show the non-SM effects on precipitation and evapotranspiration in the revised manuscript.

Figure S4. Non-SM effects on precipitation and evapotranspiration in the wet and dry seasons in CMIP6. The non-SM effects on changes in precipitation (ΔP) and evapotranspiration (ΔE) between 1980-2000 and 2080-2100 are assessed as the five-model results from LFMIP-pdLC.

Results:

“In the wet season, P-E changes in CMIP6 and LFMIP-pdLC are comparable over most land area (Fig. 3a,d), and the SM effect on P-E changes is relatively small (Fig. 3g), especially in the Northern Hemisphere, compared to P-E changes induced by other processes, such as anthropogenic climate change, in LFMIP-pdLC (collectively, we term these the non-SM effect). The drying of the wet season over subtropical regions and the Amazon is therefore mainly caused by the non-SM effect (Fig. 3a,d,g), as anthropogenic warming reduces precipitation but enhances evapotranspiration with prescribed SM in LFMIP-pdLC (Fig. S4a,d). However, the SM effect is opposite, and of similar magnitude, to the non-SM effect on P-E changes in the dry season (Fig. 3e,h). Over subtropical regions and the Amazon, the non-SM effect reduces P-E and SM in both seasons (Fig. 3d,e and Fig. 2g,h). SM drying and associated land-atmosphere processes strongly increase dry season P-E, which cancels out P-

E reductions induced by the non-SM effect, resulting in an increase in the net P-E and the wetting of the dry season (Fig. 3b,e,h).

.....

Overall, the distinct SM effects between wet and dry seasons (-0.39 ± 0.16 mm/day) are responsible for 63% of the reduced annual range of P-E, while the non-SM effect (-0.24 ± 0.25 mm/day) accounts for the remaining 37% (Fig. 3l).”

Discussion:

“Further analyses of CMIP6 land-atmosphere coupling sensitivity experiments provide evidence that the drying of the wet season is mainly caused by anthropogenic climate change, while the wetting of the dry season is driven by the negative SM feedback on P-E. We have also found that the reduced seasonality of P-E is predominantly caused by the seasonally varying SM effects on P-E, while anthropogenic climate change reduces P-E in both dry and wet seasons, and the net effect on P-E seasonality is small over the subtropics and Amazon.”

Now, to speculate a bit, what might be driving these two key effects?

For the wet season non-SM effect on P-E (3j RS downward yellow bar), I would imagine it is mostly just P reduction. That's because with SM fixed by the pdLC protocol, E won't be able to change too much with climate change (while we know that in most of the RS regions, P is indeed projected to strongly decline in the CMIP average). But you could check this speculation of mine easily.

For the dry season SM effect on P-E (3k RS upward blue bar), Fig S3b,e shows that it's dominated by E reduction, not P increase (as you again show quite clearly in Figure 4). So the SM-circulation-P feedback mechanism you invoke at e.g. 120-121 doesn't seem to be that important. Instead, it seems to be a much more straightforward story: lower annual and/or wet-season P leads to lower dry-season SM, which curtails dry-season E. Even more fundamentally, if P lowers a lot, E must also lower a lot just by water mass conservation, if we are in a water-limited region, which many of these places are. This will show up in the SM effect, rather than in the non-SM effect.

So I think the regional P-E seasonality reduction is happening for one very simple ultimate reason - *reduced total and/or wet-season P*. This both reduces wet-season P-E, and also increases dry-season P-E (makes it "less negative") by reducing the supply of water mass (soil moisture) available to supply dry-season E, while dry-season P is ~ 0 and can't change much.

If this does not make sense, think of some simple end-member examples. In a climate with no P (e.g. Saharan interior), there will be no E either. So the seasonality of P-E will be zero. On

the other hand, in a climate with moderate amount of P that falls seasonally (e.g. Sahel), P-E seasonality will be decently large because wet season P-E will be positive while dry season P-E will be negative (as plants transpire some of the water that fell in the wet season, even though no water is currently falling). Finally in a climate with very high P that falls seasonally (e.g. northeastern India or Bangladesh, or parts of subtropical South America), P-E seasonality will be gigantic, because wet season P-E will be very positive but dry season P-E will be quite negative due to the combination of \sim zero P with "wet" E from the saturated soil and groundwater. So in climates with highly seasonal P (i.e. most of your WDDW regions), the wet-season or annual P magnitude should be a very strong control on the P-E seasonality, largely via its effects on wet-season P and on dry-season E.

You do not have to completely adopt this framing, but I think you at least must mention and discuss it substantially. It would be a much simpler explanation for your results than the explanation you are giving!

Similarly, I would interpret the result at 223-224 not as increased moisture convergence, but rather as *decreased* moisture *divergence* stemming simply from the lower E. In the dry season, the key balance is that E comes out of the land surface and is diverged away by the wind (little of it is recycled as P due to lack of upward motion in dry season). So if E declines, then the moisture divergence will also have to decline. Again it is a simple mass conservation argument.

It is very interesting that this dry season moisture divergence decline is accomplished by the SH-driven decline in the strength of the subsidence (blue curve in Fig 5h; purple bar in Fig 5d) rather than a decline in the thermodynamic term, but still I think the water mass conservation has to hold. Presumably, the profile of dry season $-w$ itself (as opposed to $-\Delta w$) is sinking/subsidence both in the present and future climates, it's just weaker subsidence in the future (which would explain why P is almost unchanged, rather than increasing from the increase in $-w$ as one might expect).

Response:

We fully understand the reviewer's explanation of the reduced P-E seasonality based on water mass conservation over land. This explanation of "reduced wet season P (and P-E) \rightarrow reduced SM \rightarrow reduced E and increased P-E ($P \approx 0$) in the dry season" appears to explain the WDDW pattern in a simple way and may serve as a complement to our interpretation based on land-atmosphere coupling.

In the mechanism section, we aim to fully understand the mechanisms of the seasonally varying SM effects on P-E, and show both the SM effects on P and E, and the SM impacts on moisture

convergence ($MC \approx P-E$ at the monthly scale according to atmospheric water mass conservation) and the associated dynamic and thermodynamic processes between the wet and dry seasons (Figs. 3,4,5 and S3,S5,S6). Using modelling and empirical approaches, we first demonstrated that the SM effects on E and P-E are strong in the dry season but weak in the wet season. We also investigated the SM effects on the atmospheric moisture budget and show that SM drying modulates atmospheric vertical motion to enhance MC and increase P-E in the dry season but not in the wet season. These identified mechanisms support the reviewer's view based on the water mass conservation over land that 1) reduced wet season P decreases P-E because the SM effects on E and P are weak in the wet season; 2) reduced SM decreases dry season E but does not impact P because reduced SM enhances MC to offset the effect of E decreases on precipitation. It is worth noting that reduced SM enhances MC mainly through regulating atmospheric circulation and upward vertical motion, rather than the thermodynamic effect of reduced E and atmospheric humidity as reasoned by the reviewer.

As the reviewer's interpretation based on water mass conservation over land is essentially consistent with our explanation from the perspective of land-atmosphere coupling, we have therefore included the reviewer's interpretation in the discussion section as follows:

Discussion:

“Further analyses of CMIP6 land-atmosphere coupling sensitivity experiments provide evidence that the drying of the wet season is mainly caused by anthropogenic climate change, while the wetting of the dry season is driven by the negative SM feedback on P-E. We have also found that the reduced seasonality of P-E is predominantly caused by the seasonally varying SM effects on P-E, while anthropogenic climate change reduces P-E in both dry and wet seasons, and the net effect on P-E seasonality is small over the subtropics and Amazon. The resultant drying of the soil in turn reduces evapotranspiration and recycling of evaporated moisture for subsequent precipitation^{24,25}. Reduced evaporative cooling further amplifies land surface warming, and the associated land-ocean warming contrast strengthens surface pressure differences between ocean and land, which drives anomalous ocean-to-land moisture transport and enhances moisture convergence over land. Such an increase in SM-induced moisture convergence offsets the decrease in precipitation driven by reduced moisture recycling of evapotranspiration, resulting in a more muted precipitation response than the evapotranspiration response to SM drying and a negative SM feedback on P-E. The SM limitation on evapotranspiration and associated SM-atmosphere feedbacks, especially those related to atmospheric dynamics, are strong in the dry season but weak in the wet season, contributing to large P-E increases in the dry season and slight P-E decreases in the wet season.”

“The negative SM-(P-E) feedback caused by the SM regulation of atmospheric dynamics and moisture convergence is also expected from observational evidence that the SM limitation on evapotranspiration is strong in the dry season, when precipitation is relatively low and cannot decrease as much as evapotranspiration could in response to SM drying²⁴.”

Regarding the reviewer’s comment on moisture convergence (MC)/divergence (MD), we used MC in this study because 1) $\Delta MC \approx \Delta(P-E)$ according to atmospheric water mass conservation, therefore, our study, as well as many previous studies¹⁻³, investigates the mechanisms of P-E changes through analyzing thermodynamic and dynamic contributions of MC changes; 2) although the mean horizontal moisture flux is dominated by moisture divergences in the dry season, reduced SM may enhance MC or reduce MD in different periods/regions; 3) $\Delta MC = -\Delta MD$, reduced SM enhances MC or reduces MD are essentially the same in meaning. Therefore, we have revised the sentence to interpret the result in Fig. 5d as “increased moisture convergence (or decreased moisture divergence)”.

“Dry season evapotranspiration reductions are largely offset by increased moisture convergence (or decreased moisture divergence), resulting in only small decreases in precipitation (Fig. 5d).”

Finally, in this view, the assessment at 288-289 and 300 may be too optimistic. If what's really happening is just reduction in dry-season E due to lack of water supply, then that's clearly a "droughty" trend, not an anti-droughty trend. I.e. it would force plant productivity *declines*, not increases, seeing that dry-season E mainly represents plant water use for carbon assimilation. (And would also lead to runoff declines, since the driver is the lack of water supply [SM] in the first place, which would cause *both* E and Q to decline in the dry season - again you could easily check this using model runoff fields).

Response:

Our statements in lines 288-289 and 300 emphasize that the negative SM feedback on P-E may attenuate the potential increase of drought risk from the thermodynamic effect of climate change (or climate change induced by non-SM effect), which is directly supported by our results in Fig. 3. In the absence of the negative SM feedback on P-E, the projected droughty trend may be stronger than coupled climate projections. The importance of the SM-(P-E) feedback can be clearly identified by comparing different regions with positive and negative feedbacks. For example, we find strong wet season P-E reductions in Amazon, which are induced by both non-SM effect and the SM effect (Fig. 3d,g), as SM has a positive feedback on wet season P-E in the Amazon (Fig. 4d,j). In contrast, the wet season P-E reductions are much weaker in South Africa, as the SM effect on P-E largely offsets the P-E reduction from the non-SM effect. In the dry season, the negative SM feedback on P-E cancels out the P-E

reductions from the non-SM effect in the Amazon, South Africa, and other RS regions. Based on our assessment, we do not state that the negative SM feedback leads to an anti-droughty trend, which we also do not agree, but emphasize that the negative SM feedback on P-E may partially mitigate the droughty trend induced by climate change (or non-SM effect) in the absence of SM feedbacks.

Major plotting error:

Fig S6a-h color bar's interior is accidentally transposed in x-y space -- the colors vary *vertically* within the colorbar, instead of horizontally. This makes it impossible to figure out whether red=positive or red=negative. I inferred that red=negative, but I am not sure! This must be fixed to be readable.

Response:

We thank the reviewer for catching the plotting error. The color bar has been corrected (red for negative and blue for positive) in the revised manuscript.

Minor:

44: Ref 5 quite clearly used the phrase "intensification of the global water cycle" to mean the 1-2% increase in *global-mean P* or *global-mean E*, *not* the much more important 7% amplification of spatial/seasonal P-E pattern that they derived later in their paper and that

you're discussing here. Some writers still follow this usage from Ref 5, and some do not, leading to confusion. Therefore, for maximum clarity to all readers, I would just delete this phrase, ending up with e.g. "Global warming increases water vapor in the atmosphere. This increase is generally expected to amplify the existing..." and then cite 5,6,7,8 at the end of the second sentence.

Response:

We agree with the reviewer and have revised the sentences following the reviewer's suggestion.

"Global warming increases water vapor in the atmosphere. This increase is generally expected to amplify the existing spatial as well as seasonal patterns of P-E, leading to wet regions/seasons getting wetter, and dry regions/seasons getting drier, which is referred to as "wet get wetter, dry get drier" (WWDD) mechanism⁵⁻⁸."

69: "promotes" is a bit of an odd word choice; something like "amplifies" or "intensifies" would be more usual.

Response:

"promotes" has been revised to "amplifies" in the revised manuscript.

71: similarly "reflect" is odd here - what about just "simulate" or "project"?

Response:

"reflect" has been replaced by "project" in the revised manuscript.

81: for maximum clarity, should be "smaller percentage (~10 percentage points) of reduced seasonality regions", to make it clear that you mean e.g. 30% reducing to 20%, not 30% reducing to 27%.

Response:

We have clarified the percent change as "~10% of global land area" in comparison with results ("~20% of global land area") shown in Fig. 1.

104-106: should emphasize that this is not true in general over land (e.g. Byrne and O'Gorman 2015, Greve and Seneviratne 2015).

Response:

In the first paragraph of the section “Mechanisms of seasonal water availability changes”, we explain the thermodynamic and dynamic mechanisms of long-term P-E changes. In lines 104-106, we first introduce the thermodynamic effect of global warming on P-E, though it cannot fully explain the change in P-E over land due to the influences of atmospheric dynamic processes as well as land-atmosphere interactions (as explained later in the same paragraph). We have emphasized that the statement is a thermodynamic perspective as follows:

“Global warming, from a thermodynamic perspective, is expected to increase atmospheric water vapor and horizontal moisture transport, favoring increased P-E over wet regions of the tropics and extratropics and reduced P-E over subtropical dry regions^{5,17,18.}”

Fig 2: it would be very useful to outline the WDDW regions on the panels (e.g. with a thin black contour), so as to make it easy to quickly verify the statement at 127-129.

Fig 2: It makes no sense to plot % change in a quantity like SM that can be ~zero (that's why you have huge signals over e.g. the Sahara Desert that dominate the map). A tiny and irrelevant change in a near-zero quantity could be a huge % change. Rather, you should plot standardized change (i.e. SM change normalized by 1971-2000 SM interannual standard deviation) or dimensionless/volumetric change (i.e. change in SM divided by the depth of the soil layer over which SM is quantified). Those would each be much less prone to spurious large values in deserts than % change.

end 141-143: This last sentence made me quite confused for a while, because I thought, if SM is prescribed to be the same thing in both the historical and the future period, then how could there be any SM difference between the two periods at all? I.e. why isn't the last row of Figure 2 solid white, given this? It took me some time to realize that the last row of Figure 2 is actually the results from the *regular*, *standard* CMIP6 historical and SSP585 simulations performed by the LFMIP-pdLC-participating models (*not* the LFMIP-pdLC simulations themselves). So you must make that explicit.

Response:

As mean SM varies by a factor of ~20 geographically, we calculated the percent change in SM relative to mean SM to allow for comparison of SM changes in different regions. According to the reviewer's suggestion, we plotted the standardized change in SM normalized by the interannual standard deviation of historical SM (1970-2000) in the 30 CMIP6 models. The results also show spurious large values in the Sahara Desert (Fig. R1 below).

Fig R1. Multi-model mean seasonal changes in SM relative to the standard deviation of historical SM (1970-2000) in CMIP6.

After a careful check of the models, we found the percent change in SM is reasonable in most models, but large percent changes in SM occur in several models, such as EC-Earth3 and EC-Earth3-Veg, in which the mean SM is close to 0 mm in the Sahara Desert. In the revised Fig. 2, we have omitted the regions with spurious large values to obtain multi-model mean SM changes. Please note the large SM change in Sahel is because of the projected large increase in P-E in CMIP6 models (Fig. 1b). We have also added stippling to show the reduced seasonality (WDDW) regions in the revised figure.

Regarding the reviewer's comment on the caption of Fig. 2, the third row of Fig. 2 refers to SM changes from the CMIP6 historical and SSP585 simulations of the five models that participated in the LFMIP-pdLC experiment. We have clarified in the figure caption that the SM change in LFMIP-pdLC is zero and not shown in the figure.

Figure 2. Multi-model mean seasonal changes in soil moisture in CMIP5 and CMIP6. The percent change in soil moisture (ΔSM) is assessed between 1971-2000 (historical simulation) and 2071-2100 (RCP8.5 or SSP585 simulation) in 35 CMIP5 models (a-c) and 30 CMIP6

models (**d-f**), and between 1980-2000 (historical simulation) and 2080-2100 (SSP585 simulation) in the five CMIP6 models which participated in the LFMIP-pdLC experiment (**g-i**). In the LFMIP-pdLC simulations of the five CMIP6 models, SM is prescribed as the mean seasonal cycle of 1980-2014 for both historical and future periods (the SM change is zero and not shown in the figure). Stippling denotes reduced seasonality regions in Fig. 1e (35 CMIP5 models, first row), Fig. 1f (30 CMIP6 models, second row), and Fig. 3c (5 CMIP5 models, third row).

159: should be (Fig. S3d,e). The reader should not have to hunt for the specific panels you're referring to.

Response:

We have used “Fig. S3d,e” instead of “Fig. S3” in the revised sentence.

161-162: With the important exceptions of the Amazon and Australia - these should be noted.

Response:

We agree and have revised the sentence to express that the SM effect is relatively small in the Northern Hemisphere.

“In the wet season, P-E changes in CMIP6 and LFMIP-pdLC are comparable over most land area (Fig. 3a,d), and the SM effect on P-E changes is relatively small (Fig. 3g), especially in the Northern Hemisphere, compared to P-E changes induced by other processes, such as anthropogenic climate change, in LFMIP-pdLC (collectively, we term these the non-SM effect).”

Fig 5g,h: Red and blue colors should not be used for the arrows, since they are already used to denote ES and RS respectively. Rather, the arrows should be black so as not to confuse the viewer. The direction of the arrow alone is sufficient to explain its meaning; color (which could be confused with your ES and RS colors) isn't needed.

Response:

We have used black arrows in the revised figure.

236: I don't quite see what Fig S4 is supposed to be showing, or how to draw the conclusion here from Fig S4? You may have to walk the reader through this more. It's much easier to see in Fig. 5b,d,f.

Response:

Fig. S4 was added as suggested by the reviewer#1 in the first round of review. We agree that the original Fig. S4 is not necessary and have removed it in the revised manuscript.

238: should be (Fig. S5b,d,f).

Response:

We have cited "Fig. S5b,d,f" instead of "Fig. S5" in the revised sentence.

319-320: is there a reason that E was calculated from LH rather than taken from the CMIP output directly? The CMIP output does usually include E, as "evspsbl". The reason you did not use "evspsbl" should be explained in the text. If there's no reason, then you should directly use "evspsbl" rather than calculating from LH.

Response:

Evapotranspiration is directly available as “evspsbl” and can also be calculated from latent heat flux (“hfls”) in CMIP5 and CMIP6. Their results are consistent, as “hfls” is “evspsbl” scaled by the latent heat of vaporization. However, “evspsbl” was not available for several models from the CMIP6 website at the writing of this paper, so we finally used “hfls” which was available in more models.

We have explained the reason in the revised manuscript as follows:

“We calculated evapotranspiration from latent heat flux which was available in more CMIP6 models than evapotranspiration (“evspsbl”), and obtained precipitation minus evapotranspiration (P-E) in each model.”

390-401: The use of lag correlation here seems odd - the E (and thus P-E) should depend on the current SM, not the SM of one month ago. Did you test to see if the correlations became stronger if zero-lag was used instead of 1-month-lag? This should be mentioned in the text as well.

Response:

We used the multiple linear regression model between P-E and one-month-lagged SM because SM and P-E are strongly coupled. Therefore, it is difficult to isolate the SM feedback on P-E from the direct P-E impact on SM if zero-lag is used. To identify the SM-atmosphere feedbacks, many previous studies have used lagged correlation or regression model with lagged dependent variables^{1,4-6}. This is because of relatively long SM memory, and the SM effects on evapotranspiration and precipitation may persist for several weeks or months.

We have explained why we used one-month-lagged SM in the multiple linear regression to identify the SM feedback on P-E in the revised manuscript.

“As the SM effect on P-E may persist for weeks to months, the regression model between P-E and one-month lagged SM therefore can isolate the SM feedback on P-E from the direct P-E effect on SM²³.”

Fig S1, top row: The regions with December- and January-centered wet seasons are a bit hard to read on these maps because the plotting function attempts to squish all the other months in between the January and December regions, leading to shimmering, stringy rainbow artifacts at all the December-January (purple-to-blue) boundaries. You can especially see this in e.g. the Middle East, Central Asia, and the Canada-US border region in Fig S1e. So is there a way

to use a less-fancy plotting function that does not try to interpolate or draw contours, but instead just plots the discrete value at each pixel? That would make the December-January boundary much sharper and looking more like the remaining months' boundaries. (In matlab you would just use `pcolor` or `imagesc` instead of `contourf`, but I don't know what environment you are using here). Note that this suggestion is optional, but would help a lot if doable.

To be complete, this suggestion also applies to the second row, but for whatever reason the problem is more visually apparent in the first row (the wet season map).

Fig S1, third and fourth rows: The color scale here could really benefit from a couple of improvements. First, the value 0 should be given a completely neutral color (e.g. white), not yellow - since yellow fools the viewer into thinking it's a negative shift. Second, because shifts of -5 and of +6 mean almost the same thing in practice, it makes no sense to designate them with opposite colors. Rather, I would assign +6 a shade of intense purple (equidistant between red and blue), and then move the intense red color from -5 to -3 (and the intense blue color from +6 to +3). Finally I would interpolate between these to get the remaining months.

In short: 0=white -> -3=red -> 6=purple -> +3=blue -> 0=white, with the arrows representing interpolation in the intervening months. It may work even better if the red and blue are assigned to -4 and +4 respectively, you could try that as well (0=white -> -4=red -> 6=purple -> +4=blue -> 0=white).

Response:

After a careful check, we found the December-January boundary was caused by a resampling issue when we plotted the maps. We have corrected the resampling error in Fig. S1a,b,e,f and replotted the figure in the revised manuscript. We have also revised the color ramp In Fig. S1c,d,g,h as suggested by the reviewer.

References:

1. Zhou, S. *et al.* Soil moisture–atmosphere feedbacks mitigate declining water availability in drylands. *Nat. Clim. Chang.* **11**, 38–44 (2021).
2. Seager, R. *et al.* Model Projections of an Imminent Transition to a More Arid Climate in Southwestern North America. *Science* **316**, 1181–1184 (2007).
3. Seager, R., Naik, N. & Vecchi, G. A. Thermodynamic and Dynamic Mechanisms for Large-Scale Changes in the Hydrological Cycle in Response to Global Warming. *J. Climate* **23**, 4651–4668 (2010).

4. Tuttle, S. & Salvucci, G. Empirical evidence of contrasting soil moisture–precipitation feedbacks across the United States. *Science* **352**, 825–828 (2016).
5. Wei, J., Dickinson, R. E. & Chen, H. A Negative Soil Moisture–Precipitation Relationship and Its Causes. *J. Hydrometeor.* **9**, 1364–1376 (2008).
6. Zhang, J., Wang, W.-C. & Wei, J. Assessing land-atmosphere coupling using soil moisture from the Global Land Data Assimilation System and observational precipitation. *J. Geophys. Res.* **113**, D17119 (2008).

REVIEWERS' COMMENTS

Reviewer #2 (Remarks to the Author):

Thanks the authors for addressing my comment. I have one last comment.

It is clear that the reduced seasonality in P-E in the subtropics is mainly caused by the reduced SM (between historical and future periods), which reduces the difference in P-E between wet and dry seasons. SM-atmosphere feedbacks were found to play an important role in affecting the P-E, but the fundamental reason is the change of climate, i.e., drier soil. For this reason, the title is a little misleading for me. When talking about SM-atmosphere feedback, we usually think it is a "natural" process. But here it is forced by climate change. This is also related to my comment in last review on separating the roles of climatology change and SM-(P-E) feedback. Therefore, I think it is better to modify the title to "Diminishing seasonality of subtropical water availability dominated by climate change induced soil moisture-atmosphere feedbacks".

Reviewer #3 (Remarks to the Author):

All line numbers below refer to the *track changes* document.

This revision largely addresses my concerns. In particular it's much clearer now that the authors are saying that SM effect overall, and in particular SM effect's *seasonality*, is the main reason for the P-E seasonality reduction (rather than the less justified claim that dry-season SM effect in particular is the main reason). The role of wet-season non-SM effect (due to direct effect of anthropogenic climate change on P and E) is also more highlighted now. Also, the major figure error in the supplement was fixed, and almost all of the minor suggestions were addressed well.

However, I still think the authors are making it too complicated at 238-239, 288-294, etc. I really don't think the SM effect is 2 canceling processes (decreased ET happily/coincidentally compensated by increased moisture convergence from the circulation change resulting in little change in P). Rather it's decreased ET *causing* there to be less moisture *divergence* because the water is just not there to supply said divergence (just by mass conservation). The balance of the dry-season air column is ET in, moisture divergence out. Moisture divergence cannot happen without ET. P is negligible in the balance in the dry season. How exactly the atmosphere reduces the moisture export is quite interesting as you show (i.e. weakening of the subsident circulation as SH increases) but it shouldn't distract from the fundamental story, which is less ET supply, therefore less moisture export by conservation of mass, and overall a *drier* atmosphere (not a wetter one) due to drying SM effects. It makes little sense from my view to claim that drying SM and decreasing ET is "wetting" the system.

You could check this yourself by looking at e.g. how the SM effect changes the dry-season relative and/or specific humidity. Also, as suggested before, you could check the SM effect on the dry-season runoff changes, and/or the dry-season vegetation (GPP, LAI) changes. I am sure you will find all of these effects are negative, not positive.

Again you do not have to change your core interpretation - your interpretation is quite interesting as well. But I think at least the much simpler ET-supply-driven perspective on the change of the moisture convergence/divergence must be *mentioned* and discussed in the paper's text (not just in the response to me) as an alternative to your core interpretation.

Also, at end 145 - beginning 148: I don't think you need this long descriptive sentence "In the LFMIP-pdLC simulations... not shown in the figure". To me, this is the sentence that is causing all the confusion (because the reader is wondering if it's not shown in the figure, then what *is* shown in the figure?) It will be much clearer without the sentence. They will be able to look up

what is the LFMIP-pdLC experiment in the body text. The exact nature of the LFMIP-pdLC is not important for understanding this figure, since this figure does not show LFMIP-pdLC results. Therefore, it's confusing to go into a long explanation of what is LFMIP-pdLC in this caption.

In addition, you should insert the word "also" just before "participated" on 145 to make it extra clear that you are not showing LFMIP-pdLC itself here, instead you're just showing regular CMIP6 simulations of those 5 models.

Finally at 149: (5 CMIP5 models, third row) should be (5 CMIP6 models, third row).

Reviewer #2 (Remarks to the Author):

Thanks the authors for addressing my comment. I have one last comment.

It is clear that the reduced seasonality in P-E in the subtropics is mainly caused by the reduced SM (between historical and future periods), which reduces the difference in P-E between wet and dry seasons. SM-atmosphere feedbacks were found to play an important role in affecting the P-E, but the fundamental reason is the change of climate, i.e., drier soil. For this reason, the title is a little misleading for me. When talking about SM-atmosphere feedback, we usually think it is a “natural” process. But here it is forced by climate change. This is also related to my comment in last review on separating the roles of climatology change and SM-(P-E) feedback. Therefore, I think it is better to modify the title to “Diminishing seasonality of subtropical water availability dominated by climate change induced soil moisture-atmosphere feedbacks”.

Response:

We thank the reviewer for the suggestion. We agree that the diminished seasonality of P-E is caused by both climate change induced SM reductions and seasonally-varying SM-atmosphere feedbacks. However, the suggested title is confusing as our attribution analysis demonstrates that the diminished seasonality of P-E is dominated by the SM effect compared to the non-SM effect, i.e., anthropogenic climate change. To highlight climate change as suggested by the reviewer, we have revised the title to “Diminishing seasonality of subtropical water availability in a warmer world dominated by soil moisture-atmosphere feedbacks”.

Reviewer #3 (Remarks to the Author):

All line numbers below refer to the *track changes* document.

This revision largely addresses my concerns. In particular it's much clearer now that the authors are saying that SM effect overall, and in particular SM effect's *seasonality*, is the main reason for the P-E seasonality reduction (rather than the less justified claim that dry-season SM effect in particular is the main reason). The role of wet-season non-SM effect (due to direct effect of anthropogenic climate change on P and E) is also more highlighted now. Also, the major figure error in the supplement was fixed, and almost all of the minor suggestions were addressed well.

However, I still think the authors are making it too complicated at 238-239, 288-294, etc. I really don't think the SM effect is 2 canceling processes (decreased ET happily/coincidentally compensated by increased moisture convergence from the circulation change resulting in little

change in P). Rather it's decreased ET *causing* there to be less moisture *divergence* because the water is just not there to supply said divergence (just by mass conservation). The balance of the dry-season air column is ET in, moisture divergence out. Moisture divergence cannot happen without ET. P is negligible in the balance in the dry season. How exactly the atmosphere reduces the moisture export is quite interesting as you show (i.e. weakening of the subsident circulation as SH increases) but it shouldn't distract from the fundamental story, which is less ET supply, therefore less moisture export by conservation of mass, and overall a *drier* atmosphere (not a wetter one) due to drying SM effects. It makes little sense from my view to claim that drying SM and decreasing ET is "wetting" the system.

You could check this yourself by looking at e.g. how the SM effect changes the dry-season relative and/or specific humidity. Also, as suggested before, you could check the SM effect on the dry-season runoff changes, and/or the dry-season vegetation (GPP, LAI) changes. I am sure you will find all of these effects are negative, not positive.

Response:

We thank the reviewer for the suggestion. In this study, we examined the “wet get drier, dry get wetter” pattern from the perspective of P-E. We agree that the dry season is not wetter in terms of reduced SM and ET, and the wetting of dry season refers to the increase in P-E in the dry season rather than the wetting of the land surface system. We have also highlighted this in the revised discussion as follows:

“It is worth noting that while soil moisture-atmosphere feedbacks lead to increases in surface water availability in the dry season, reduced soil moisture itself and associated declining evapotranspiration also indicate an overall drying trend of the land surface system driven by climate change over subtropical dry regions and the Amazon.”

Again you do not have to change your core interpretation - your interpretation is quite interesting as well. But I think at least the much simpler ET-supply-driven perspective on the change of the moisture convergence/divergence must be *mentioned* and discussed in the paper's text (not just in the response to me) as an alternative to your core interpretation.

Response:

We have included the ET-supply-driven perspective in the results and discussion sections as follows:

“Dry season evapotranspiration reductions are largely offset by increased moisture convergence (or decreased moisture divergence due to reduced supply of water vapor through evapotranspiration), resulting in only small decreases in precipitation (Fig. 5d).”

“On the other hand, reduced evapotranspiration leads to reduced terrestrial supply of water vapor for moisture divergence and therefore curbs the reduction of P-E in the dry season.”

Also, at end 145 - beginning 148: I don't think you need this long descriptive sentence "In the LFMIP-pdLC simulations... not shown in the figure". To me, this is the sentence that is causing all the confusion (because the reader is wondering if it's not shown in the figure, then what *is* shown in the figure?) It will be much clearer without the sentence. They will be able to look up what is the LFMIP-pdLC experiment in the body text. The exact nature of the LFMIP-pdLC is not important for understanding this figure, since this figure does not show LFMIP-pdLC results. Therefore, it's confusing to go into a long explanation of what is LFMIP-pdLC in this caption.

Response:

The sentence has been removed.

In addition, you should insert the word "also" just before "participated" on 145 to make it extra clear that you are not showing LFMIP-pdLC itself here, instead you're just showing regular CMIP6 simulations of those 5 models.

Finally at 149: (5 CMIP5 models, third row) should be (5 CMIP6 models, third row).

Response:

The figure caption has been revised accordingly.